# Fip1 is a multivalent interaction scaffold for processing factors in human mRNA 3′ end biogenesis

Lena Maria Muckenfuss, Anabel Carmen Migenda Herranz, Franziska Maria Boneberg, Marcello Clerici, Martin Jinek*

Department of Biochemistry, University of Zurich, Zurich, Switzerland

**Abstract** 3′ end formation of most eukaryotic mRNAs is dependent on the assembly of a ~1.5 MDa multiprotein complex, that catalyzes the coupled reaction of pre-mRNA cleavage and polyadenylation. In mammals, the cleavage and polyadenylation specificity factor (CPSF) constitutes the core of the 3′ end processing machinery onto which the remaining factors, including cleavage stimulation factor (CstF) and poly(A) polymerase (PAP), assemble. These interactions are mediated by Fip1, a CPSF subunit characterized by high degree of intrinsic disorder. Here, we report two crystal structures revealing the interactions of human Fip1 (hFip1) with CPSF30 and CstF77. We demonstrate that CPSF contains two copies of hFip1, each binding to the zinc finger (ZF) domains 4 and 5 of CPSF30. Using polyadenylation assays we show that the two hFip1 copies are functionally redundant in recruiting one copy of PAP, thereby increasing the processivity of RNA polyadenylation. We further show that the interaction between hFip1 and CstF77 is mediated via a short motif in the N-terminal 'acidic' region of hFip1. In turn, CstF77 competitively inhibits CPSF-dependent PAP recruitment and 3′ polyadenylation. Taken together, these results provide a structural basis for the multivalent scaffolding and regulatory functions of hFip1 in 3′ end processing.

*For correspondence:
jinek@bioc.uzh.ch

## Editor's evaluation

This study explores the structural and biochemical basis for Fip1 interactions within the cleavage and polyadenylation machinery – notably with CPSF30 and CstF77. Overall, the significance of the study is that it provides valuable mechanistic insight into the function of Fip1 in the cleavage and polyadenylation machinery. The data presented in the paper are compelling and the authors use a combination of structural biology and biochemistry to present their case. This study will be of interest to those focusing on mRNA biosynthesis and the biophysical properties of RNA binding proteins.

## Introduction

3′ end polyadenylation is a fundamental process in eukaryotic messenger RNA (mRNA) biogenesis, essential for the maturation of non-histone precursor mRNAs (pre-mRNAs) prior to their export into the cytoplasm. Poly(A) tails possess key functions in mRNA metabolism, governing mRNA export, translational efficiency, and stability (*Nicholson and Pasquinelli, 2019*; *Passmore and Coller, 2022*). Furthermore, alternative mRNA polyadenylation (APA) constitutes a key mechanism of gene expression control through dynamic regulation of polyadenylation site selection in pre-mRNA transcripts (*Di Giammartino et al., 2011*; *Tian and Manley, 2016*). Accordingly, defects in polyadenylation are linked to human diseases such as cancer, β-thalessemia, diabetes, or systemic lupus (*Hollerer et al., 2014*; *Gruber and Zavolan, 2019*; *Dharmalingam et al., 2022*). mRNA 3′ end biogenesis occurs by a two-step mechanism comprising endonucleolytic cleavage of the

pre-mRNA transcript by the cleavage and polyadenylation specificity factor (CPSF) complex and subsequent polyadenylation of the free 3' end by the poly(A) polymerase (PAP). In human cells, the process is dependent on the controlled assembly of several protein factors on the pre-mRNA, including CPSF, RBBP6, cleavage stimulation factor (CstF), as well as mammalian cleavage factors I and II (CF Im and CFIIm, respectively), and PAP (*Zhao et al., 1999*; *Xiang et al., 2014*; *Kumar et al., 2019*; *Boreikaite et al., 2022*; *Schmidt et al., 2022*). Most of these protein factors are highly conserved between mammals and yeast, underlining the fundamental nature of this process (*Xiang et al., 2014*). The cleavage site, typically downstream of a CA dinucleotide, is defined by the polyadenylation signal (PAS), a conserved hexanucleotide motif (predominantly AAUAAA) located approximately 10–30 nucleotides upstream (*Proudfoot and Brownlee, 1976*; *Proudfoot, 2011*).

The PAS is specifically recognized by CPSF (*Chan et al., 2014*; *Schönemann et al., 2014*; *Clerici et al., 2018*; *Sun et al., 2018*), which consists of two functional modules: the mammalian polyadenylation specificity factor (mPSF) comprising subunits CPSF160, WDR33, CPSF30, and hFip1 (for human factor interacting with poly(A) polymerase 1) (*Bienroth et al., 1991*; *Murthy and Manley, 1992*; *Kaufmann et al., 2004*; *Shi et al., 2009*), and the mammalian cleavage factor (mCF) containing the endonuclease CPSF73 (*Mandel et al., 2006*), CPSF100 as well as Symplekin (*Sullivan et al., 2009*). RBBP6 associates with mCF and is essential for pre-mRNA cleavage (*Di Giammartino et al., 2014*; *Boreikaite et al., 2022*; *Schmidt et al., 2022*). Within mPSF, the CPSF160–WDR33 subcomplex forms a rigid scaffold (*Clerici et al., 2017*) that interacts with CPSF30 (*Clerici et al., 2018*; *Sun et al., 2018*; *Zhang et al., 2019*) and the CPSF100 subunit of mCF (*Zhang et al., 2019*). CPSF30 contains five C3H1-type zinc finger (ZF) domains and a C-terminal zinc knuckle domain which is absent in yeast homolog Yth1 (*Barabino et al., 1997*) and not required for mPSF complex assembly (*Clerici et al., 2017*). The ZF1 domain is necessary and sufficient for binding to the CPSF160–WDR33 heterodimer, while ZF2 and ZF3 together with WDR33 mediate recognition of the AAUAAA PAS hexamer motif (*Clerici et al., 2018*; *Sun et al., 2018*). ZF4 and ZF5 domains interact with hFip1 (*Barabino et al., 2000*; *Hamilton and Tong, 2020*). hFip1 is an important regulator of APA that contributes to cleavage site selection through its interaction with CFIm via its C-terminal arginine/serine-rich (RS) domain (*Zhu et al., 2018*) and additionally by binding to U-rich regions in the pre-mRNA via an arginine-rich C-terminal region (*Kaufmann et al., 2004*), thereby specifically promoting polyadenylation of mRNA substrates with U-rich sequences preceding the AAUAAA hexanucleotide (*Lackford et al., 2014*). Both the RS domain and the arginine-rich region are absent in yeast Fip1 and the alternatively spliced isoform 4 of hFip1.

In previously determined cryo-EM structures of the yeast CPF and human mPSF complexes, ZF4 and ZF5 remained unresolved (*Casañal et al., 2017*; *Clerici et al., 2018*; *Sun et al., 2018*), indicating conformational flexibility with respect to the rigid mPSF core. Recently, a crystal structure of human CPSF30 ZF4–5 domains in complex with hFip1 has been determined (*Hamilton and Tong, 2020*) and complementary NMR studies of the yeast Fip1 homolog *Kumar et al., 2021* have shed light on the molecular details of the CPSF30–Fip1 interaction and revealed considerable structural dynamics of Fip1 in the context of the 3' processing machinery.

Mammalian CstF is a dimer of trimers comprising CstF77, CstF64, and CstF50 subunits (*Takagaki et al., 1990*; *Yang et al., 2018*). It is recruited to the pre-mRNA by U- and G/U-rich sequences located downstream of the cleavage site (*Takagaki and Manley, 1997*) that are recognized by CstF64 (*Takagaki et al., 1992*; *MacDonald et al., 1994*). Through stabilization of CPSF on the pre-mRNA, CstF plays an important role in PAS recognition and is essential for pre-mRNA cleavage (*Takagaki et al., 1990*; *Boreikaite et al., 2022*; *Schmidt et al., 2022*). Dimerization of CstF is mediated by the CstF77 HAT (half-a-tetratricopeptide repeat) domain homodimer (*Bai et al., 2007*), and further stabilized by CstF50 (*Yang et al., 2018*). The CstF77 homodimer has an arch-like shape and interacts asymmetrically with CPSF, contacting the CPSF160–WDR33 mPSF scaffold via only one side of the arch (*Zhang et al., 2019*).

Fip1 interacts with PAP and tethers it to CPSF bound near the nascent 3' end of the cleaved pre-mRNA, which is required for its processive polyadenylation (*Preker et al., 1995*; *Helmling et al., 2001*; *Kaufmann et al., 2004*; *Meinke et al., 2008*; *Ezeokonkwo et al., 2011*). Besides CPSF30 and PAP, biochemical and cellular studies have implicated Fip1 in interactions with other proteins including CPSF160, CstF77 (*Preker et al., 1995*; *Kaufmann et al., 2004*), WDR33 (*Ohnacker et al.,*

*2000*; *Clerici et al., 2017*), Symplekin (*Ghazy et al., 2009*), and CF Im (*Venkataraman et al., 2005*). However, the molecular details of these interactions have not yet been revealed.

Here, we report structural and biochemical analysis of the interactions of hFip1 with CPSF30, PAP, and CstF77 within the human 3′ polyadenylation machinery. While confirming previous structural data (*Hamilton and Tong, 2020*), we notably show that mPSF contains two hFip1 copies, yet recruits only one PAP molecule at a time. The presence of two PAP-binding sites in mPSF contributes to the processivity of 3′ polyadenylation. Furthermore, we show that hFip1 interacts with CstF77 through a conserved helix in its N-terminal 'acidic' region and reveal that CstF77 competes with PAP for hFip1 binding, which attenuates polyadenylation efficiency. These results deepen our understanding of hFip1 as a key interaction partner for 3′ end processing factors, facilitating or regulating their spatio-temporal assembly on the pre-mRNA, and establish a framework for further mechanistic studies of hFip1 interactions and CstF-mediated regulation of mRNA 3′ end biogenesis.

## Results

### Structural basis for the human hFip1–CPSF30 interaction

The ZF4 and ZF5 domains of CPSF30 are necessary and sufficient for the interaction with the conserved central domain of hFip1 (hereafter referred to as hFip1$^{CD}$) (*Clerici et al., 2017*; *Hamilton and Tong, 2020*). Yet these domains could not be resolved in previously determined cryo-EM reconstructions of the human mPSF (*Clerici et al., 2018*; *Sun et al., 2018*), indicating that they are likely flexibly tethered to the mPSF core. To gain insights into the CPSF30–hFip1 interaction, we determined a crystal structure of a CPSF30 fragment spanning ZF4 and ZF5 domains (CPSF30$^{ZF4–ZF5}$, residues 118–178) in complex with hFip1$^{CD}$ (residues 138–180 of hFip1 isoform 4) at a resolution of 2.2 Å (*Figure 1B*). The structure reveals that hFip1$^{CD}$ binds CPSF30 in a 2:1 stoichiometry, with one hFip1$^{CD}$ molecule (hFip1$^{CD}$a) binding predominantly to ZF4 and the other (hFip1$^{CD}$b) to ZF5. While the overall conformation of the hFip1–CPSF30 complex is highly similar to that of a recent crystal structure of human hFip1–CPSF30 (*Hamilton and Tong, 2020*), with a root-mean-square deviation of 0.82 Å over the whole complex, in this study, 21 additional residues of hFip1 could be resolved in hFip1$^{CD}$b due to extended crystallization construct boundaries, with 16 additional residues at the N-terminus (residues 130–145, isoform 4) and 5 additional residues at the C-terminus (residues 177–181, isoform 4). Structural superpositions reveal that hFip1$^{CD}$a and hFip1$^{CD}$b bind to the same surfaces of ZF4 and ZF5 domains with a root-mean-square deviation of 0.87 Å over 54 aligned residues (*Figure 1C*). Moreover, superpositions with CPSF30 ZF2 and ZF3 domains reveal that the interaction surfaces on ZF4 and ZF5 are located on the opposite side of the ZF fold relative to the PAS RNA-binding surfaces of ZF2 and ZF3 (*Figure 1C*). ZF2 and ZF3 interactions with the RNA are mainly mediated by π–π stacking of aromatic side chains with nucleobases and supplemented by protein mainchain hydrogen bond interactions (*Clerici et al., 2018*; *Sun et al., 2018*). Although the aromatic residues are conserved in ZF4 and ZF5 (*Figure 1—figure supplement 1*), RNA binding is likely precluded by the presence of proline residues at key mainchain hydrogen-binding positions. The hFip1$^{CD}$a–CPSF30 interaction surface (803 Å$^2$) is almost twice as large as the hFip1$^{CD}$b–CPSF30 interface (478 Å$^2$) because hFip1$^{CD}$a binds at the ZF4–ZF5 junction and has additional contacts with ZF5. hFip1$^{CD}$a and hFip1$^{CD}$b also contact each other directly (215 Å$^2$). ZF4 interaction with hFip1$^{CD}$a is mediated by a hydrophobic interface centered on Lys127$^{CPSF30}$ and Phe131$^{CPSF30}$ and supported by additional salt-bridge contact involving Arg144$^{CPSF30}$ and Asn159$^{hFip1}$ (*Figure 1D*). In turn, the interaction of ZF5 with hFip1$^{CD}$b is mainly mediated by Tyr151$^{CPSF30}$ and Phe155$^{CPSF30}$ and supported by a salt-bridge contact between Arg168$^{CPSF30}$ and Asp159$^{hFip1}$ (*Figure 1E*). As Fip1 is conformationally dynamic in isolation (*Meinke et al., 2008*; *Ezeokonkwo et al., 2011*; *Kumar et al., 2021*), CPSF30-binding results in structural ordering of the CD region. Interactions with both ZF4 and ZF5 are mediated by a hydrophobic patch in hFip1$^{CD}$ comprising the aromatic side chains of Trp150$^{hFip1}$, Phe161$^{hFip1}$, and Trp170$^{hFip1}$ (*Figure 1D, E*).

To validate our structural observations, we initially mutated ZF4 and ZF5 interaction surface residues in CPSF30$^{ZF4–ZF5}$ and tested the interactions of the mutant proteins with hFip1$^{CD}$ in a pull-down assay (*Figure 1—figure supplement 2A*). Individual substitutions of Tyr127$^{CPSF30}$, Tyr151$^{CPSF30}$, or Phe155$^{CPSF30}$ with glutamate resulted in substantial reduction of hFip1$^{CD}$ binding, while simultaneous mutation of both ZF4 and ZF5 residues resulted in loss of hFip1 binding, in agreement with our structural observations. In hFip1$^{CD}$, substitution of aromatic residues with glutamate in the hydrophobic interaction

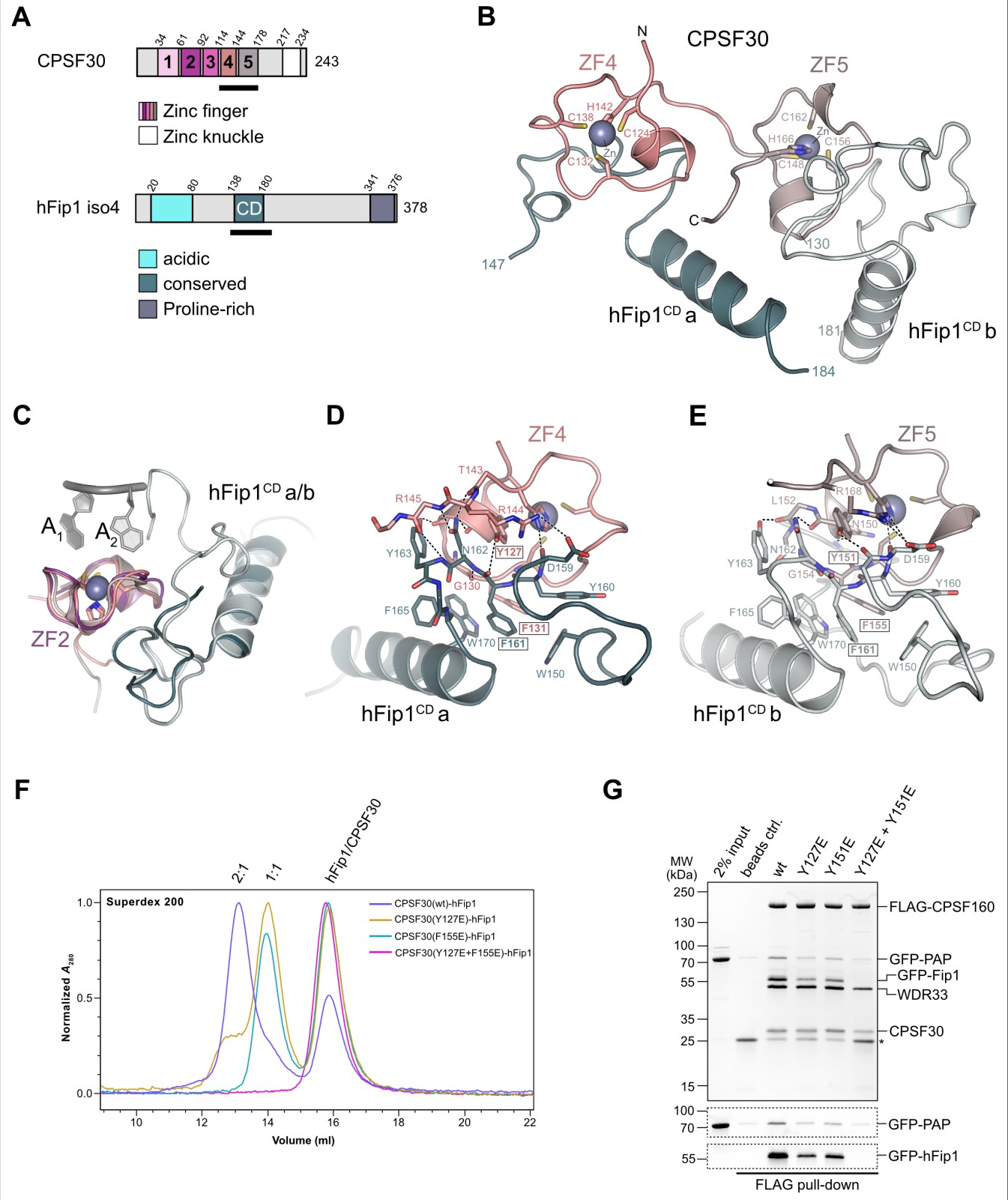

**Figure 1.** hFip1 interacts with CPSF30 with 2:1 stoichiometry. (**A**) Schematic representation of the domain architecture of CPSF30 and hFip1. CPSF30 consists of five zinc finger (ZF) domains and a zinc knuckle domain. hFip1 isoform 4 comprises acidic, conserved, and proline-rich regions but lacks the RE/D region interacting with CF $I_m$, as well as the R-rich region, which has been shown to bind U-rich RNA in hFip1 isoform 1 (*Kaufmann et al., 2004*). (**B**) Cartoon representation of the crystal structure of CPSF30ZF4–ZF5 in complex with two hFip1 fragments comprising the conserved domain (CD). (**C**)

*Figure 1 continued on next page*

*Figure 1 continued*

Superposition of CPSF30 ZF2 domain in complex with PAS RNA onto ZF4 and ZF5. (**D**) Detailed interaction interface of hFip1$^{CD}$ with CPSF30 ZF4. (**E**) Detailed interaction interface of hFip1$^{CD}$ with CPSF30 ZF5. (**F**) Size-exclusion chromatography coupled to multiangle static light scattering (SEC-MALS) chromatogram of MBP-CPSF30$^{ZF4-ZF5}$ selective hFip1-binding mutants for stoichiometry analysis with GFP-hFip1. (**G**) In vitro pull-down analysis of FLAG-epitope-tagged mPSF comprising wild-type CPSF30 and its selective hFip1-binding mutants with GFP-PAP. Asterisk indicates anti-FLAG M2 antibody light chain. GFP-hFip1 and GFP-PAP are also visualized with in-gel GFP fluorescence (bottom).

The online version of this article includes the following source data and figure supplement(s) for figure 1:

**Source data 1.** Raw gel image for *Figure 1*, panel G.

**Source data 2.** Raw gel image for *Figure 1*, panel G.

**Figure supplement 1.** Sequence alignment of CPSF30 zinc finger domains.

**Figure supplement 2.** Analysis of the hFip1–CPSF30 interaction using structure-guided point mutants.

**Figure supplement 2—source data 1.** Raw gel image for *Figure 1—figure supplement 2*, panel A.

**Figure supplement 2—source data 2.** Raw gel image for *Figure 1—figure supplement 2*, panel A.

**Figure supplement 2—source data 3.** Raw gel image for *Figure 1—figure supplement 2*, panel B.

**Figure supplement 2—source data 4.** Raw gel image for *Figure 1—figure supplement 2*, panel B.

patch either substantially reduced (Trp150$^{hFip1}$ and Trp170$^{hFip1}$) or completely disrupted (Phe161$^{hFip1}$) the hFip1$^{CD}$–CPSF30$^{ZF4-ZF5}$ interaction (*Figure 1—figure supplement 2B*). We subsequently performed size-exclusion chromatography coupled to multiangle static light scattering (SEC-MALS) to analyze the stoichiometry of hFip1$^{CD}$–CPSF30$^{ZF4-ZF5}$ complexes. hFip1$^{CD}$ and wild-type CPSF30$^{ZF4-ZF5}$ formed a 2:1 complex. In contrast, CPSF30$^{ZF4-ZF5}$ proteins containing Y127E$^{CPSF30}$ or F155E$^{CPSF30}$ mutations formed a 1:1 complex with hFip1$^{CD}$, while simultaneous mutation of both residues resulted in complete loss of binding (*Figure 1F*). Together, these results confirm that human CPSF30 has two independently functional hFip1-binding sites, one on ZF4 and the other on ZF5, each recruiting one copy of hFip1.

## Functional redundancy of hFip1–CPSF30 interactions in human CPSF

To probe the functional significance of the dual CPSF30–hFip1 interaction interfaces in the context of human CPSF, we coexpressed wild-type or mutant CPSF30 together with hFip1, WDR33 and FLAG epitope-tagged CPSF160 in baculovirus-infected insect cells, and performed tandem affinity purifications during which purified recombinant catalytic domain of human PAP (residues 1–504) was added in trans after the second affinity purification step. hFip1 copurified with mPSF containing wild-type CPSF30, and PAP was efficiently coprecipitated (*Figure 1G*). Expression of CPSF30 ZF4 or ZF5 mutants (Y127E or Y151E, respectively) resulted in reduced recovery of both hFip1 and PAP relative to the other mPSF components (*Figure 1G*), consistent with the reduced stoichiometry of the CPSF30–hFip1 interaction observed in vitro (*Figure 1F*). In turn, expression of a CPSF30 construct containing mutations in both the ZF4- and ZF5-binding sites resulted in the loss of hFip1 from mPSF, which was thus unable to interact with PAP (*Figure 1G*). Together, these results indicate that either hFip1-binding site in CPSF30 can contribute to the integrity of mPSF in vivo and both sites are capable of recruiting hFip1 and consequently PAP. Notably, the expression levels of mPSF mutant complexes incapable of binding hFip1 (Y127E/Y151$^{CPSF30}$) were substantially reduced, consistent with the role of hFip1 in stabilizing the CPSF30 ZF fold (*Kumar et al., 2021*).

We next assessed the requirement of the hFip1–CPSF30 interactions for RNA 3′ polyadenylation using an in vitro polyadenylation assay. Incubation of a model RNA substrate with purified wild-type mPSF (*Figure 2—figure supplement 1A*) and PAP resulted in processive addition of ~60 adenosine nucleotides, which was dependent on the presence of ATP and the AAUAAA hexameric PAS in the RNA (*Figure 2A*). The efficiency of 3′ polyadenylation was reduced upon incubation of the substrate with mPSF complexes containing CPSF30 ZF4 or ZF5 mutants capable of binding only one copy of hFip1 (*Figure 2A*). No polyadenylation was observed upon incubation with mPSF containing the CPSF30 ZF4/ZF5 double mutant (*Figure 2A*), consistent with the loss of hFip1 and PAP recruitment (*Figure 1G*), and polyadenylation could not be rescued by the addition of recombinant hFip1 in trans (*Figure 2A*). Collectively, these observations indicate that both hFip1-binding sites in CPSF30 contribute to the efficiency of RNA 3′ polyadenylation, suggesting that the presence of two hFip1 copies, and thus two PAP recruitment sites, in mPSF is required for highly efficient, processive 3′

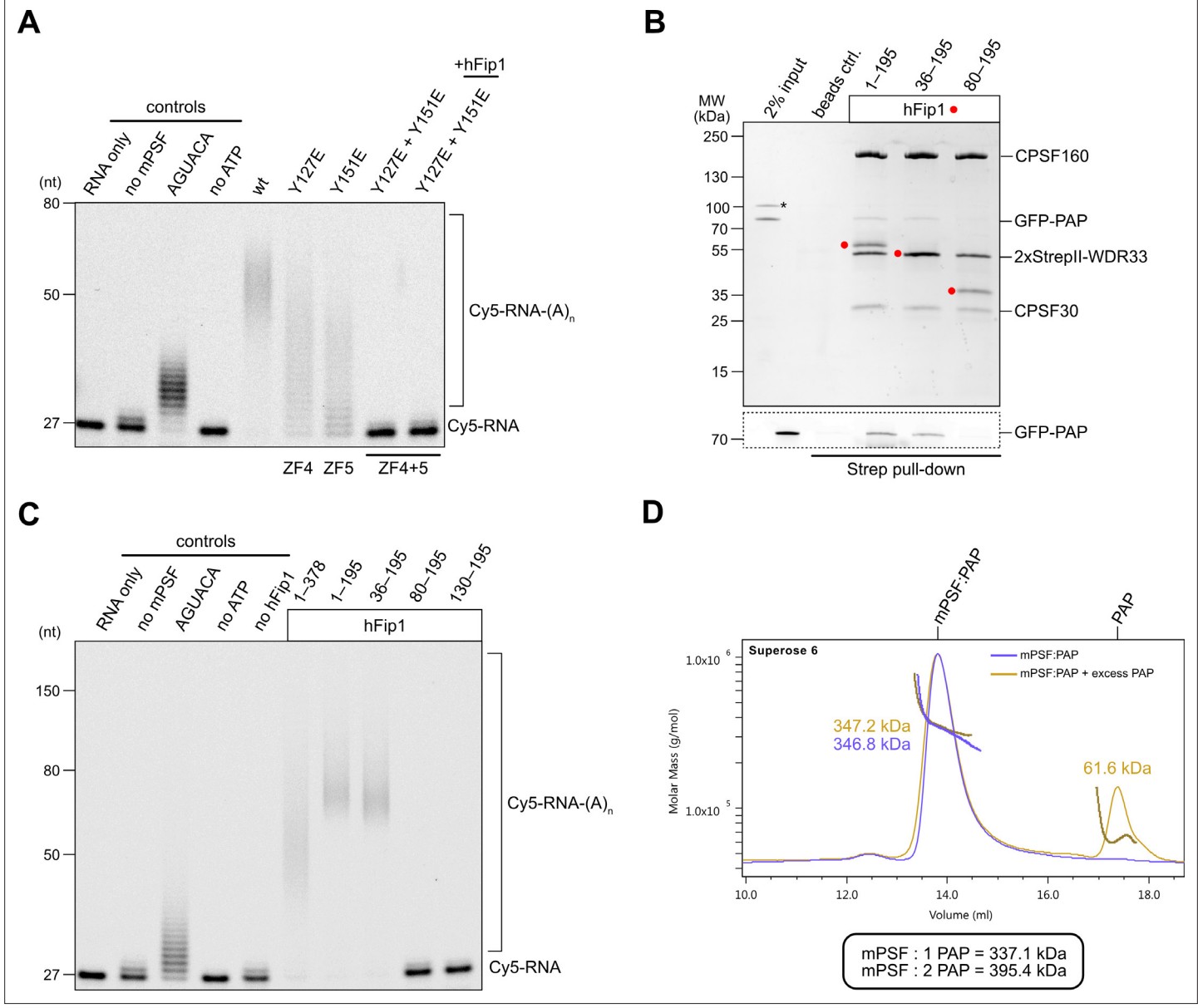

**Figure 2.** hFip1 directly recruits poly(A) polymerase. (**A**) Polyadenylation activity assay of mPSF complexes containing wild-type and mutant CPSF30 proteins as well as hFip1 added in trans (rightmost lane) using a Cy5-labeled PAS-containing RNA substrate. An RNA substrate lacking the canonical AAUAAA PAS hexanucleotide is denoted by its substitute sequence, AGUACA. Polyadenylated RNA products are indicated as RNA-(A)$_n$. (**B**) Pull-down analysis of immobilized StrepII-tagged mPSF complexes comprising N-terminal truncations of hFip1 with GFP-PAP. GFP-PAP is visualized by in-gel GFP fluorescence (bottom). Asterisk denotes contaminating protein. (**C**) Polyadenylation activity assay of mPSF complexes containing hFip1 truncations. (**D**) Size-exclusion chromatography coupled to multiangle static light scattering (SEC-MALS) analysis of reconstituted mPSF:PAP:RNA complexes and in the absence (purple) or presence of excess PAP (yellow). Theoretical molecular masses of 1:1 and 1:2 mPSF:PAP complexes are indicated.

The online version of this article includes the following source data and figure supplement(s) for figure 2:

**Source data 1.** Raw gel image for *Figure 2*, panel A.

**Source data 2.** Raw gel image for *Figure 2*, panel A.

**Source data 3.** Raw gel image for *Figure 2*, panel B.

**Source data 4.** Raw gel image for *Figure 2*, panel B.

**Source data 5.** Raw gel image for *Figure 2*, panel C.

**Source data 6.** Raw gel image for *Figure 2*, panel C.

**Figure supplement 1.** Analysis of hFip1 regions required for PAP recruitment.

*Figure 2 continued on next page*

*Figure 2 continued*

**Figure supplement 1—source data 1.** Raw gel image for *Figure 2—figure supplement 1*, panel A.
**Figure supplement 1—source data 2.** Raw gel image for *Figure 2—figure supplement 1*, panel A.
**Figure supplement 1—source data 3.** Raw gel image for *Figure 2—figure supplement 1*, panel B.

polyadenylation. However, neither hFip1-binding site is strictly necessary for RNA 3′ polyadenylation, suggesting their functional redundancy.

## PAP recruitment occurs via hFip1 N-terminal region

In *S. cerevisiae*, a poorly conserved peptide motif in the N-terminal region of Fip1 directly interacts with the poly(A) polymerase Pap1 (*Meinke et al., 2008*). Similarly, the N-terminal region of human hFip1, upstream of the CD, is required for PAP interaction (*Kaufmann et al., 2004*) but the precise PAP interaction site in human hFip1 has not been identified. To this end, we tested the interaction of green fluorescent protein (GFP)-tagged PAP with purified mPSF complexes containing truncated hFip1 fragments in an in vitro pull-down experiment. PAP was detectably, albeit weakly, coprecipitated by mPSF containing a hFip1 fragment spanning both the N-terminal and CD regions (residues 1–195) as well as by mPSF containing an N-terminally truncated hFip1 (residues 36–195) (*Figure 2B*). However, further N-terminal truncation of hFip1 resulted in the loss of PAP binding, indicating that a region spanning residues 36–80 in human hFip1 is required for PAP interaction (*Figure 2B*). An additional pull-down experiment using recombinant PAP and glutathione-*S*-transferase (GST)-fused hFip1 fragments revealed that although the hFip1 region comprising residues 36–80 was required for PAP interaction, it was not sufficient (*Figure 2—figure supplement 1B*). This suggests that additional parts of hFip1 contribute to PAP binding.

We subsequently tested the activity of mPSF complexes containing N- or C-terminally truncated hFip1 in the polyadenylation assay. In agreement with the interaction data, mPSF complexes containing hFip1 fragments spanning residues 1–195 or 36–190 were able to support efficient RNA 3′ polyadenylation (*Figure 2C*), whereas mPSF complexes containing hFip1 fragments comprising residues 80–195 or 130–195 were not. Together, these results indicate that hFip1 residues 36–80 are required for the recruitment of PAP to effect mPSF-dependent 3′ polyadenylation. Interestingly, we also observed that polyadenylation levels were reduced with mPSF containing full-length hFip1 (residues 1–378, isoform 4), as compared to mPSF containing C-terminally truncated hFip1 (residues 1–195), suggesting that the C-terminal region of hFip1, which is proline-rich and predicted to be intrinsically disordered, negatively modulates the processivity of mPSF-dependent 3′ polyadenylation.

## CPSF recruits only one copy of PAP

Prior studies have indicated that the polymerase module of endogenous yeast CPF comprises up to two copies of Pap1 (*Casañal et al., 2017*). Furthermore, a complex comprising human CPSF30 ZF4 and ZF5 domains and two hFip1 molecules is capable of simultaneous interaction with two PAP molecules in vitro (*Hamilton and Tong, 2020*). To determine whether this also occurs in the context of human mPSF, we analyzed the mPSF–PAP interaction by SEC-MALS. Despite only weakly interacting in pull-down analysis, at high PAP concentrations (40 μM), mPSF and PAP formed a stable complex that could be purified by SEC. Analysis of this complex using SEC-MALS revealed an apparent molecular mass of 347 kDa, closely matching the predicted molecular mass of a complex containing two hFip1 molecules and one PAP (337 kDa) (*Figure 2D*). Addition of excess PAP to the prepurified mPSF–PAP sample did not cause peak broadening; furthermore, no change in the detected apparent molecular mass could be observed. These results suggest that mPSF predominantly associates with only one PAP molecule at a time, despite the presence of two copies of hFip1 in the complex.

## The N-terminal region of hFip1 interacts with CstF77

In analogy with the yeast polyadenylation machinery, human Fip1 has previously been shown to interact with CstF77 (*Preker et al., 1995*; *Kaufmann et al., 2004*). To validate these observations and identify the interaction determinants in hFip1, we performed a pull-down experiment with GST-tagged hFip1 fragments and maltose-binding protein (MBP)-tagged fragment of CstF77 comprising the HAT domain (residues 21–549). The very N-terminal region of hFip1 spanning residues 1–35 was

necessary and sufficient for the interaction with the CstF77 HAT domain (*Figure 3A*). Notably, this region is dispensable for the interaction of hFip1 with PAP and for RNA 3′ polyadenylation (*Figure 2B, C*).

To shed light on the hFip1–CstF77 interaction, we subsequently reconstituted a complex comprising the hFip1$^{1–35}$ fragment with a truncated construct of the CstF77 HAT domain (residues 241–549) and determined its X-ray crystallographic structure at a resolution of 2.7 Å. The structure reveals that hFip1 binds to a conserved positively charged patch located on the convex surface of the CstF77 HAT domain arch (*Figure 3B*, *Figure 3—figure supplement 1A, B*). Within the hFip1$^{1–35}$ fragment, only the evolutionarily conserved residues 20–27 were ordered, adopting an alpha-helical conformation (*Figure 3C, D*). Interaction of hFip1$^{1–35}$ with CstF77 involves salt-bridge contacts of Glu22$^{hFip1}$ and Glu23$^{hFip1}$ with Arg402$^{CstF77}$, and hydrophobic contacts involving Leu26$^{hFip1}$ and Tyr27$^{hFip1}$ with Phe398$^{CstF77}$, Val428$^{CstF77}$, Ile432$^{CstF77}$, and Leu435$^{CstF77}$. Additionally, the Tyr27$^{hFip1}$ side chain interacts with Arg395$^{CstF77}$ via π–π stacking. Corroborating these structural observations, simultaneous substitutions of Glu22$^{hFip1}$ and Glu23$^{hFip1}$, or Trp25$^{hFip1}$, Leu26$^{hFip1}$ and Tyr27$^{hFip1}$ with alanine disrupted the hFip1$^{1–35}$–CstF77$^{21–549}$ interaction in a pull-down experiment, whereas alanine substitution of Trp25$^{hFip1}$ alone did not have an effect (*Figure 3E*). In turn, mutation of the positively charged interaction surface in CstF77 (Arg395, Arg402, and Lys431 mutated to alanines) abolished the interaction with hFip1$^{1–35}$ (*Figure 3E*).

A previously determined cryo-EM reconstruction of the human mPSF–CstF77 complex revealed that the interaction of the CstF77 HAT domain dimer with mPSF is primarily mediated by extensive contacts with WDR33 and CPSF160 (*Zhang et al., 2019*). Upon close inspection, the cryo-EM map from this dataset (EMDB entry EMD-20861) exhibits residual densities on both CstF77 protomers that could be attributed to the binding of two hFip1 molecules via their N-terminal regions (*Figure 3H*). This observation indicates that CstF77 is capable of binding two hFip1 copies when bound to mPSF. We subsequently tested the contribution of hFip1$^{1–35}$ to the mPSF–CstF77 interaction in a pull-down experiment using MBP-tagged CstF77 and mPSF complexes containing truncated hFip1 fragments. Although all mPSF complexes were capable of binding CstF77, reduced levels of CstF77 coprecipitation were observed with mPSF containing N-terminally truncated hFip1 that lacked the CstF77 interacting region (*Figure 3—figure supplement 2A*). Furthermore, to exclude that the hFip1-binding site is obstructed upon assembly of CstF77 into a holo-CstF complex, we tested interaction of GST-tagged hFip1$^{1–35}$ in a pull-down experiment using with purified holo-CstF comprising full-length CstF77, CstF50, and CstF64$^{1–198}$. Comparable amounts of holo-CstF and MBP-CstF77 were precipitated by GST-hFip1$^{1–35}$ indicating that the Fip1 interaction interface in CstF77 remains exposed upon CstF complex assembly (*Figure 3—figure supplement 3A*). Taken together, these results suggest that direct interactions between hFip1 and CstF77 contribute to the assembly of the CPSF–CstF complex during mRNA 3′ end biogenesis.

## CstF inhibits polyadenylation

As CstF77 and PAP bind to nonoverlapping, yet adjacent, sites in hFip1, CstF77 binding could nevertheless preclude PAP recruitment due to steric crowding. To probe this, we carried out a pull-down experiment with GST-tagged hFip1 and mixtures of MBP-tagged CstF77 and GFP-tagged PAP at varying molar ratios. In the presence of excess CstF77, PAP binding was considerably reduced, indicating that CstF77 competes with PAP for binding to hFip1 (*Figure 4A*). Consistent with this, CPSF-dependent RNA 3′ polyadenylation was substantially reduced in the presence CstF77, suggesting that CstF77 inhibits 3′ polyadenylation via interaction with hFip1 (*Figure 4B*). The inhibitory effect of CstF77 was reduced either when mPSF lacked the N-terminal CstF77 interaction site in hFip1 (*Figure 4B*) or when CstF77 (CstF$^{mut}$) was incapable of interaction with the N-terminal region of hFip1 (*Figure 4—figure supplement 1A*), indicating that the inhibitory effect of CstF77 is in part dependent on its interaction with hFip1. In both cases, addition of excess CstF77 led to a reduction of RNA 3′ polyadenylation rate, although not to the same extent (*Figure 4B*, *Figure 4—figure supplement 1A*). Furthermore, inhibition of 3′ polyadenylation was also observed upon addition of a holo-CstF complex, suggesting that the inhibitory effect of CstF77 is maintained when it is assembled within CStF (*Figure 4C*). To exclude the possibility that the observed CstF77-dependent reduction of RNA 3′ polyadenylation rate might be due to the close proximity of the PAS and the 3′ terminal CA dinucleotide in the SV40 mRNA-based substrate (11 nt), we tested an alternative RNA substrate (adenoviral L3

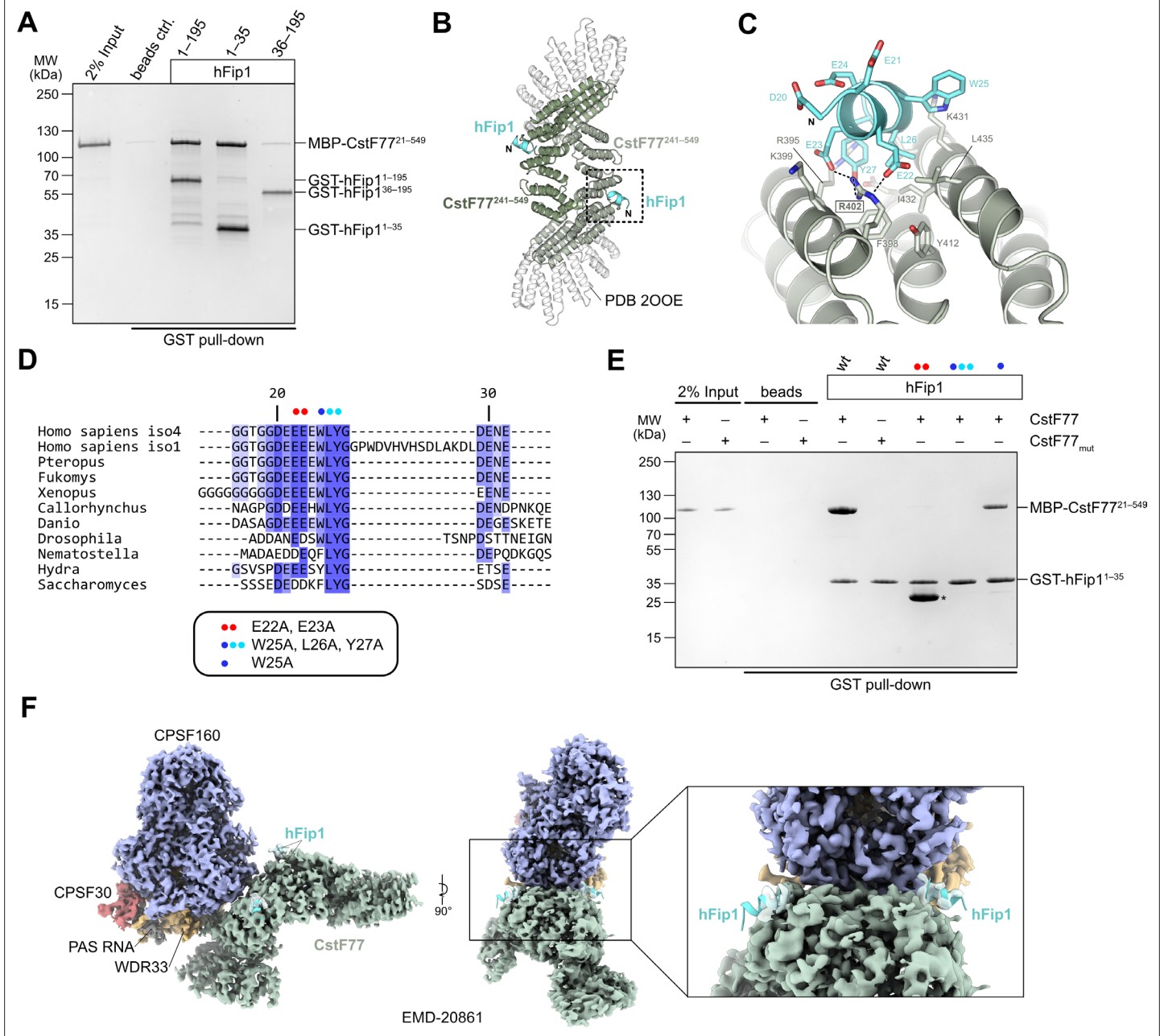

**Figure 3.** hFip1 interacts with CstF77 through a conserved motif within its N-terminal acidic domain. (**A**) Pull-down analysis of immobilized GST-hFip1 fragments with MBP-CstF77²¹⁻⁵⁴⁹. (**B**) Cartoon representation of the crystal structure of the CstF77²⁴¹⁻⁵⁴⁹–hFip1¹⁻³⁵ complex, superimposed onto the structure of murine CstF77 (white, PDB ID: 2OOE). (**C**) Zoomed-in view of the hFip1–CstF interaction interface. (**D**) Multiple sequence alignment of the N-terminal region of Fip1 orthologs. (**E**) Pull-down analysis of immobilized wild-type and mutant GST-hFip1¹⁻³⁵ proteins with MBP-CstF77²¹⁻⁵⁴⁹ and MBP-CstF77mut (R395A/R402A/K431A). Asterisk indicates contaminating free GST protein. (**F**) 3D cryo-EM density map (EMD-20861) of the human CPSF160–WDR33–CPSF30–PAS RNA–CstF77 complex (*Zhang et al., 2019*), displayed at contour level 0.015 and color coded according to the corresponding atomic protein model (PDB ID 6URO). The hFip1–CstF77 crystal structure from this study was superimposed onto the atomic model of CstF77, and atomic model of hFip1 is shown (cyan). Inset shows a zoomed-in view of unassigned density that matches hFip1.

The online version of this article includes the following source data and figure supplement(s) for figure 3:

**Source data 1.** Raw gel image for *Figure 3*, panel A.

**Source data 2.** Raw gel image for *Figure 3*, panel E.

**Figure supplement 1.** hFip1 binds to a conserved positively charged patch on CstF77.

**Figure supplement 2.** The N-terminal region of hFip1 contributes to mPSF–CstF77 interaction.

*Figure 3 continued on next page*

*Figure 3 continued*

**Figure supplement 2—source data 1.** Raw gel image for *Figure 3—figure supplement 2*, panel A.

**Figure supplement 3.** The N-terminal region of hFip1 interacts with CstF complex.

**Figure supplement 3—source data 1.** Raw gel image for *Figure 3—figure supplement 3*, panel A.

PAS-containing mRNA) in which the PAS and the cleavage site are separated by 20 nt and observed a similar degree of inhibition (*Figure 4—figure supplement 1B*), suggesting that the inhibition is likely not a consequence of steric crowding due to the proximity of the PAS and the 3′ end. Altogether, these results suggest that CstF inhibits RNA 3′ polyadenylation via CstF77, and that CstF77-mediated inhibition is in part dependent on its interaction with hFip1.

## Discussion

Despite extensive efforts to obtain structural insights into the molecular organization and regulation of the eukaryotic mRNA 3′ end processing machinery, high-resolution structural information has so far only been obtained for stable subassemblies composed of structurally rigid subunits (*Casañal et al., 2017*; *Clerici et al., 2017*; *Clerici et al., 2018*; *Sun et al., 2018*; *Hill et al., 2019*; *Zhang et al., 2019*; *Hamilton and Tong, 2020*; *Kumar et al., 2021*). Although hFip1 is an integral component of the CSPF complex, specifically its mPSF module, it has not been structurally visualized in this context owing to its intrinsically disordered nature (*Meinke et al., 2008*).

In our study, we reveal the molecular basis for the interactions of human hFip1 with both CPSF30, PAP, and CstF77. While confirming the 2:1 binding stoichiometry of the hFip1:CPSF30 interaction in isolation (*Hamilton and Tong, 2020*; *Kumar et al., 2021*), we expand this finding to the CPSF complex, confirming that its mPSF module assembles with two hFip1 copies in cells and demonstrating that both the ZF4 and ZF5 domains in CPSF30 are capable of binding hFip1 independently. Using polyadenylation assays we show that the two hFip1 copies are functionally redundant in recruiting PAP to the mPSF, which increases the processivity of RNA 3′ polyadenylation. As the recruitment of PAP to the 3′ end of the cleaved pre-mRNA is prerequisite for its processivity (*Ezeokonkwo et al., 2011*), while PAP only weakly associates with the mPSF, the presence of two hFip1 copies thus likely increases the 3′ polyadenylation efficiency polyadenylation by increasing the local PAP concentration. While recent studies of human CPSF30 and its yeast homolog Yth1 reported higher binding affinity for the Fip1:ZF4 interaction as compared to Fip1:ZF5 (*Hamilton and Tong, 2020*; *Kumar et al., 2021*), we show that polyadenylation efficiency is reduced equally independent of which hFip1 interaction site (ZF4 or ZF5) is impaired. This indicates that PAP recruitment by mPSF is the limiting factor in 3′ polyadenylation.

Although the yeast Fip1–Pap1 interaction has been extensively characterized, the Pap1 interaction motif in Fip1, as observed in the crystal structure of the complex (*Meinke et al., 2008*), is poorly conserved in human Fip1 (*Figure 4—figure supplement 2A*), only partially mapping to hFip1 residues 80–86 (hFip1 isoform 4). Recent analysis of the yeast Fip1–Pap1 interaction using nuclear magnetic resonance, however, showed that additional residues located N-terminally of the Pap1-binding motif are involved in Pap1 interaction (*Kumar et al., 2021*). This aspartate-rich acidic region is well conserved in hFip1, corresponding to residues 58–79 (hFip1 isoform 4). In agreement with this, we show that an additional N-terminal segment in hFip1 spanning residues 36–80 and including the aspartate-rich acidic motif is required but not sufficient for PAP binding. This suggests that the hFip1–PAP interaction mode closely resembles that of yeast Fip1–Pap1, despite low sequence conservation of the respective interaction motifs in yeast Fip1 and hFip1.

Notably, our biophysical analysis of the human mPSF–PAP interaction reveals that despite the presence of two hFip1 copies, only one copy of PAP is stably recruited by mPSF. This finding is unexpected and in contrast to the previous observation that two copies of the PAP catalytic domain are stably bound by the CPSF30–hFip1 subcomplex in isolation (*Hamilton and Tong, 2020*). Moreover, up to two Pap1 copies have been detected in the polymerase module of yeast CPF by native mass spectrometry (*Casañal et al., 2017*). It is important to note that although our experimental data indicate that only one PAP associates with mPSF, this does not exclude the possibility that mPSF binds a second copy of PAP with a low affinity. Nevertheless, we speculate that the observed 1:1 mPSF–PAP complex likely reflects the predominant state under physiological conditions. As the efficiency of the 3′ polyadenylation reaction depends on PAP recruitment, the presence of two PAP-binding sites in

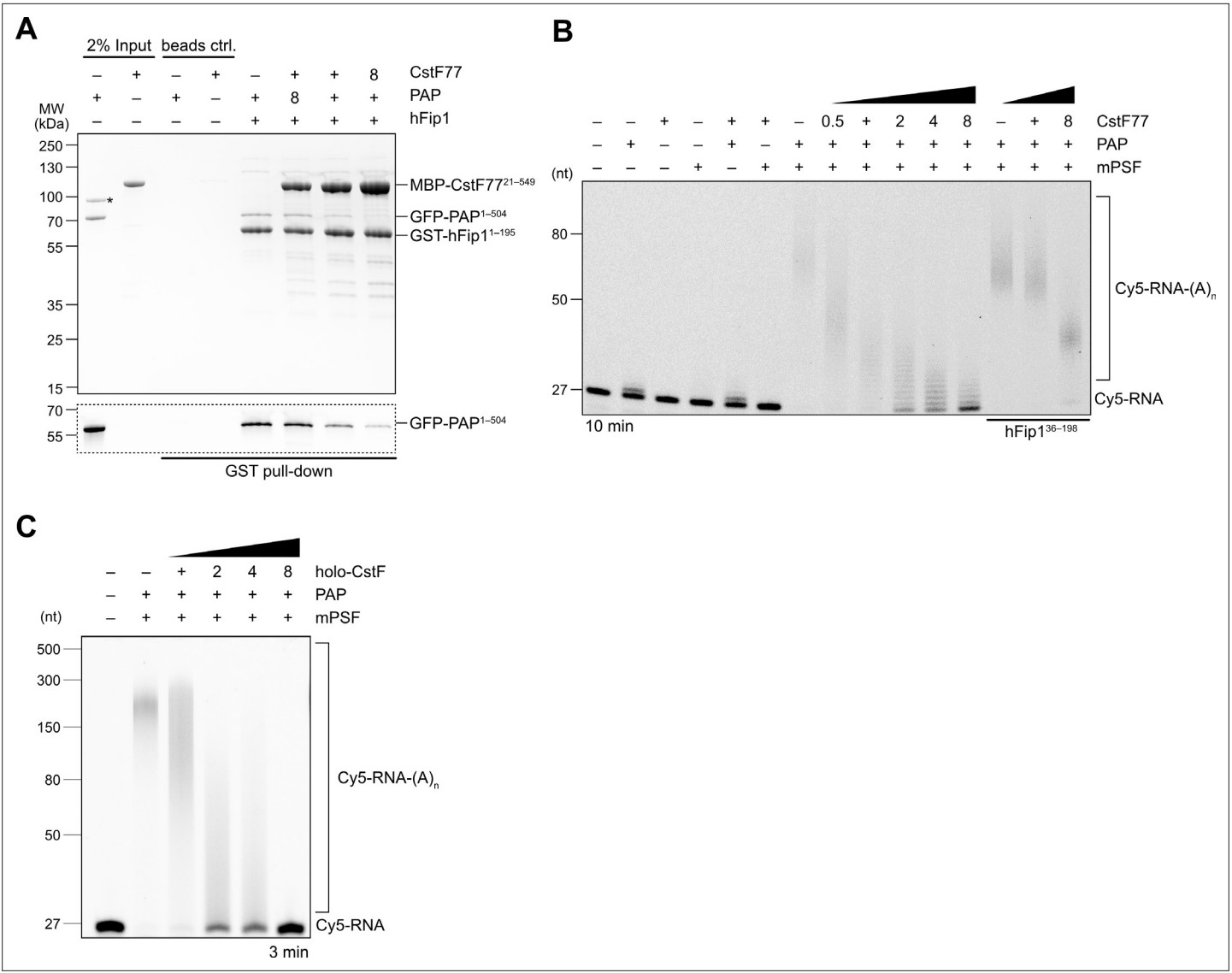

**Figure 4.** CstF77 competitively inhibits 3′ polyadenylation. (**A**) Pull-down analysis of immobilized GST-hFip1$^{1–195}$ with varying molar ratios of GFP-PAP and MBP-CstF77$^{21–549}$. GFP-PAP is visualized by in-gel GFP fluorescence (bottom). Asterisk denotes contaminating protein. (**B**) Polyadenylation activity assay of mPSF complexes containing full-length hFip1 and N-terminally truncated hFip1 (hFip1$^{36–195}$) in the presence of varying molar ratios of CstF77. Polyadenylated RNA products are indicated as RNA-(A)$_n$. (**C**) Polyadenylation activity assay of mPSF in the presence of varying molar ratios of holo-CstF complex.

The online version of this article includes the following source data and figure supplement(s) for figure 4:

**Source data 1.** Raw gel image for *Figure 4*, panel A.

**Source data 2.** Raw gel image for *Figure 4*, panel A.

**Source data 3.** Raw gel image for *Figure 4*, panel B.

**Source data 4.** Raw gel image for *Figure 4*, panel B.

**Source data 5.** Raw gel image for *Figure 4*, panel C; same as *Figure 2—source data 5*.

**Figure supplement 1.** CstF77 reduces RNA 3′ polyadenylation rate.

**Figure supplement 1—source data 1.** Raw gel image for *Figure 4—figure supplement 1*, panel A.

**Figure supplement 1—source data 2.** Raw gel image for *Figure 4—figure supplement 1*, panel A.

**Figure supplement 1—source data 3.** Raw gel image for *Figure 4—figure supplement 1*, panel B.

**Figure supplement 2.** Pap1 interaction motif in Fip1 orthologs is poorly conserved.

mPSF thus might not serve to simultaneously populate both sites with one PAP molecule each but to increase the probability of PAP recruitment to enhance polyadenylation efficiency. Furthermore, the presence of two hFip1 copies might instead be required for mPSF integrity and its interactions with CstF. In this context, it is conceivable that the binding of two PAP molecules to mPSF bound to a substrate RNA is precluded due to molecular crowding or steric hindrance, particularly considering that the two Fip1 molecules make nonidentical interactions with mPSF, and the two Fip1-binding sites in the CstF77 homodimer are not equivalent when CstF is bound to mPSF.

The interaction between CPSF and CstF has previously been shown to involve direct contacts between the CstF77 homodimer and an extensive interface provided by the CPSF160 and WDR33 subunits of CPSF (*Zhang et al., 2019*), yet CstF also interacts with CPSF via hFip1 (*Kaufmann et al., 2004*). Our crystal structure of the hFip1–CstF77 subcomplex reveals that a hFip1 binds via conserved motif within the N-terminal 'acidic' region to the convex arch of the CstF77 HAT domain on both protomers in the CstF77 homodimer, resulting in a 2:2 stoichiometry. By reanalysis of previously reported cryo-EM data (*Zhang et al., 2019*), we reveal that this interaction mode is preserved in the context of the mPSF–CstF complex. Strikingly, both CstF77 and holo-CstF inhibit 3' polyadenylation in vitro. Moreover, CstF77-mediated inhibition is partially dependent of its interaction with hFip1, which likely reflects the dual interaction mode of CStF with mPSF. Accordingly, the hFip1–PAP and hFip1–CstF77 interactions appear to be competitive, possibly as a result of the proximity of the PAP and CstF77 interaction sites within the hFip1 N-terminus. As CStF is strictly required for CPSF73-dependent pre-mRNA cleavage while PAP might not be (*Boreikaite et al., 2022*), these results imply that the CPSF–CstF interaction is disrupted or undergoes a remodeling after cleavage to enable PAP recruitment to the cleaved pre-mRNA and subsequent 3' polyadenylation.

To perform the coupled reaction steps of cleavage and polyadenylation, the 3' end processing machinery likely undergoes a sequence of conformational and compositional rearrangements as polyadenylation site recognition by the mPSF module of CPSF and activation by RBBP6 triggers CstF-dependent cleavage by the mCF, after which the nascent 3' end needs to be made accessible to PAP

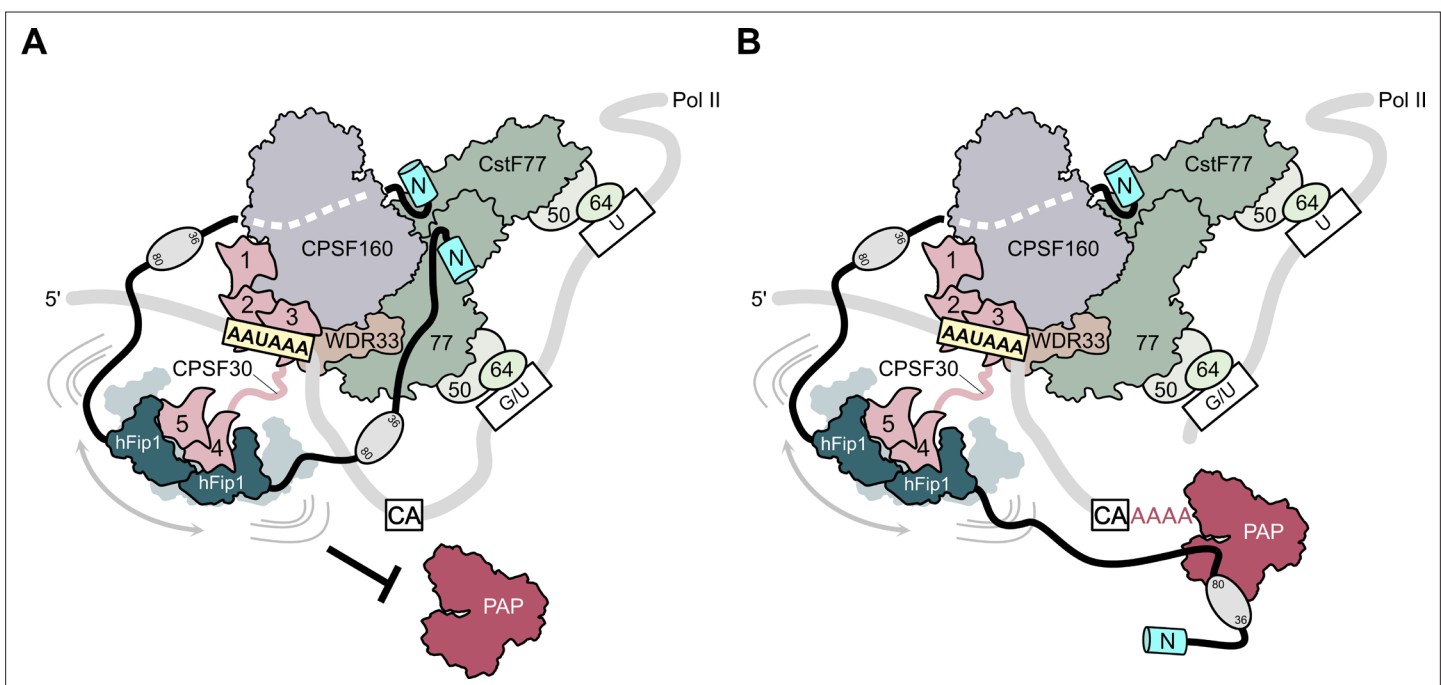

**Figure 5.** Model of CPSF-mediated pre-mRNA cleavage and polyadenylation and CstF77-dependent inhibition of polyadenylation. (**A**) Prior to pre-mRNA cleavage, PAP recruitment is inhibited by CStF, in part due to competitive interactions of CstF77 and hFip1 (left). (**B**) Upon pre-mRNA cleavage, structural remodeling of the CPSF–CstF complex enables hFip1 to recruit PAP to the nascent 3' end of the mRNA and consequently stimulates polyadenylation. *Figure 1—figure supplement 1*: sequence alignment of CPSF30 zinc finger domains. (**A**) Sequence alignment of human CPSF30 zinc finger domains. Residues responsible for RNA interactions (in ZF2/ZF3) or hFip1 interaction (in ZF4/ZF5) are highlighted and the nature of their interaction color coded. ZF4/ZF5 domains contain proline residues (yellow) at positions corresponding to critical main chain hydrogen bonding interactions in ZF2/ZF3.

for subsequent poly(A) synthesis (*Boreikaite et al., 2022*; *Schmidt et al., 2022*). Based on our structural and biochemical findings, we propose a model in which hFip1 helps coordinate the two steps of 3′ end processing. Initially, the two hFip1 molecules present in mPSF facilitate the assembly of CPSF and CstF on the pre-mRNA via the interactions of their N-terminal motifs with CstF77 (*Figure 5A*). In part, these interactions also preclude PAP recruitment until the pre-mRNA has been cleaved and a free 3′ end has been generated. Upon endonucleolytic cleavage of the pre-mRNA by CPSF73, remodeling of the 3′ end processing machinery, possibly enabled by the dissociation of the downstream pre-RNA cleavage product and concomitant release of CstF, reduces steric constraints around the nascent 3′ end and exposes PAP interaction sites in hFip1, enabling PAP recruitment to initiate processive 3′ polyadenylation of the cleaved pre-mRNA (*Figure 5B*). The conformational and compositional transitions required for accessing the nascent 3′ end are orchestrated by hFip1 and facilitated by its flexible attachment to mPSF via CPSF30, as well as by its intrinsic conformational dynamics (*Meinke et al., 2008*; *Ezeokonkwo et al., 2011*; *Kumar et al., 2021*). The presence of two hFip1 molecules in the 3′ end processing complex promotes efficient PAP recruitment and contributes to the processivity of 3′ end polyadenylation. Our model implies that PAP might be recruited to the pre-mRNA only after the cleavage step, which is supported by recent findings reported by Boreikaite et al., demonstrating that the presence of PAP is not required for endonucleolytic cleavage by mCF, but is contradicted by the study of Schmidt et al., which reported that PAP is required for pre-mRNA cleavage. The functional role of PAP in pre-mRNA cleavage thus remains unclear, necessitating further studies. Finally, the functional role of hFip1 as a major interaction platform for 3′ end processing factors is also important in the context of the its well-documented role in regulating alternative polyadenylation (*Lackford et al., 2014*) as it suggests that the interactions with processing factors might be further modulated by direct interactions of Fip1 with U-rich sequences in the pre-mRNA.

In sum, these results advance our understanding of hFip1 as a multivalent interaction hub in mRNA 3′ end processing and unravel a novel aspect of polyadenylation regulation by CstF. Through interspacing binding sites for processing factors with intrinsically disordered, low-complexity sequences hFip1 can achieve the required degree of conformational freedom to accommodate the remodeling of the 3′ end processing machinery and ensure correct spatiotemporal regulation of the processing factors at the nascent mRNA 3′ end. The molecular basis of these transitions, however, awaits further structural and functional investigations.

## Materials and methods
### Protein expression and purification
#### Protein expression vectors
To generate polypromoter plasmids for *E. coli* expression using ligation-independent cloning (LIC) analogous to MacroLab Series 438, MacroLab Series 2 vectors were modified by inserting whole expression cassettes as gBlocks (IDT) into 2B-T with Gibson assembly carrying the necessary modifications. The resulting vectors denoted 16-B (for His$_6$-TEV-tagged protein expression), 16 M (for His$_6$-MBP-TEV-tagged protein expression), and 16M_ΔHis (for MBP-TEV-tagged protein expression) have two *PmeI* restriction sites flanking the T7 expression cassette, an internal *SspI* site for target gene insertion, and a *SwaI* site downstream of the T7 terminator for biobricks-type assembly using LIC. Gene assembly proceeds following the Series 438 vectors assembly protocol (*Gradia et al., 2017*).

#### Cloning for expression in *E. coli*
Constructs encoding for CPSF30 isoform 3 (Uniprot O95639-3), hFip1 isoform 4 (Uniprot Q6UN15-4), poly(A) polymerase alpha (Uniprot P51003-1), and CstF77 (Uniprot Q12996-1) were cloned into LIC expression vectors 1B (gift from Scott Gradia, Addgene plasmid #29653), 1 M (Addgene plasmid #29656), 2 G-T (Addgene plasmid #29707), 2GFP-T (Addgene plasmid #29716), and cotransformation vector 13 S-A (Addgene plasmid 48323), respectively. DNA encoding for hFip1[130–195] was first cloned into 2 G-T, PCR amplified starting from the GST-tag and inserted into 13 S-A using LIC cloning. Point mutations in CPSF30, hFip1, PAP, and CstF77 were introduced by obtaining linear DNA fragments (GeneArt Strings, Thermo Fisher) encoding for the desired construct with LIC overhangs and cloned into the respective expression vectors according to *Supplementary file 1*.

## Cloning for expression in Sf9 cells

DNA encoding human CPSF160 (Uniprot Q10570), WDR33 (Uniprot Q9C0J8-1), CPSF30 isoform 3, and hFip1 isoform 4 were cloned into MacroBac Series 438 cloning system vectors (*Gradia et al., 2017*) according to *Supplementary file 1*. Subcloning of three- or four-subunit mPSF complexes into a single baculovirus transfer plasmid was performed following the MacroBac protocol (*Gradia et al., 2017*). For FLAG-tagged mPSF complexes, subcloning was performed using the biGBac protocol (*Weissmann et al., 2016*).

## Expression and purification of CPSF30 and hFip1 for SEC-MALS

His$_6$-MBP-TEV-CPSF30$^{1-243}$ wt and mutants (Y127E, F155E, and Y127E/F155E) were expressed overnight in *E. coli* BL21 star (DE3) cells and His$_6$-GFP-TEV-hFip1$^{1-195}$ in *E. coli* Rosetta2 (DE3) cells, respectively, at 18°C by addition of isopropyl thio-β-galactoside (IPTG) to a final concentration of 0.5 mM at OD$_{600}$ of about 0.6–0.8. Cells were resuspended in buffer A (25 mM Tris–HCl pH 7.5, 200 mM NaCl) supplemented with 0.5 mM Tris (2-carboxyethyl)phosphine (TCEP), 1 µM Pepstatin, and 400 µM 4-(2-aminoethyl)benzenesulfonyl fluoride hydrochloride (AEBSF) protease inhibitor followed by lysis via sonication. Lysate was cleared by centrifugation (20 min, 20,000 × *g*, 4°C) and clarified lysate was purified on Ni-NTA agarose resin (QIAGEN) eluted with buffer A supplemented with 0.5 mM TCEP and 200 mM imidazole. The protein was further purified by size-exclusion chromatography on a Superdex 75 (Cytiva) column, eluting with buffer A supplemented with 1 mM dithiothreitol (DTT). Eluting peak fractions were concentrated in centrifugal filter (Amicon Ultra-15, MWCO 30 kDa, Merck Millipore), flash frozen, and stored at −80°C.

## MBP-CPSF30 proteins for pull-down analysis

His$_6$-MBP-TEV-CPSF30$^{1-243}$ mutants were expressed and purified as described for SEC-MALS. For CPSF30 point mutants Y151E, Y127E/Y151E, Y127E/Y155E, a high salt wash (25 mM Tris–HCl pH 7.5, 1 M NaCl) during Ni-IMAC purification was included prior to elution with buffer A buffer supplemented with 200 mM imidazole.

## PAP expression and purification

His$_6$-MBP-TEV-PAP$^{1-504}$ was expressed in *E. coli* Rosetta2 (DE3) cells overnight at 18°C by induction with 0.5 mM IPTG at OD$_{600}$ of about 0.6–0.8. Cells were lysed by sonication in 20 mM Tris–HCl pH 8.0, 500 mM NaCl, 5 mM imidazole, 0.5 mM TCEP supplemented with 0.1 µM Pepstatin, and 400 µM AEBSF protease inhibitor. Lysate was cleared by centrifugation (20 min, 20,000 × *g*, 4°C) and cleared lysate was subjected to Ni-NTA resin (QIAGEN), washed and protein eluted with buffer B (20 mM Tris–HCl pH 8.0, 200 mM NaCl) supplemented with 200 mM imidazole. Eluted protein was further purified on MBPTrap HP (Cytiva), eluting in buffer B supplemented with 10 mM maltose. The eluted protein fractions were injected onto a Superdex 200 column (Cytiva) equilibrated in buffer B supplemented with 1 mM DTT. Tag was cleaved off the protein with His$_6$-MBP-TEV protease, and the cleavged tags including protease removed from protein sample using a MBPTrap HP (Cytiva). For use in pull-down analysis, the tag was not cleaved from His$_6$-MBP-TEV-PAP$^{1-504}$ wt and mutant (R395A, R402A, and K431A) after size-exclusion chromatography. Purified protein was concentrated in centrifugal filter (Amicon Ultra-15, MWCO 50 kDa, Merck Millipore), aliquoted, flash frozen, and stored at −80°C.

## Expression and purification of GFP-PAP for pull-down analysis

His$_6$-GFP-TEV-PAP$^{1-504}$ was expressed overnight in *E. coli* Rosetta2 (DE3) cells at 18°C by induction with 0.5 mM IPTG at OD$_{600}$ of about 0.6–0.8. Cells were lysed by high-pressure cell disruption at 25 kpsi in buffer B supplemented with 5 mM imidazole, 0.5 mM TCEP, 0.1 µM Pepstatin, and 400 µM AEBSF protease inhibitor. Lysate was cleared by centrifugation (20 min, 20,000 × *g*, 4°C) and clarified lysate was subjected to Ni-NTA resin (QIAGEN), washed and protein eluted with buffer B supplemented with 250 mM imidazole. Eluted protein was further purified on Superdex 200 column (Cytiva) equilibrated in 20 mM Tris–HCl pH 8.0, 150 mM NaCl, and 0.5 mM TCEP. Purified protein was concentrated in centrifugal filter (Amicon Ultra-15, MWCO 50 kDa, Merck Millipore), aliquoted, flash frozen, and stored at −80°C.

## Expression of mPSF complexes

For expression of mPSF complexes in *Sf9* cells (Thermo Fisher Scientific, cat. no. 11496015; cell line was authenticated and tested for mycoplasma contamination by the manufacturer, no further validation was done by the authors), recombinant baculoviruses were generated according to the Bac-to-Bac Baculovirus expression system (Invitrogen). 2 ml of P3 virus were used to infect 1 l of *Sf9* insect cells at a density of $1.1 \times 10^6$ ml$^{-1}$. Cells were harvested 72 hr postinfection.

## Purification of mPSF complexes for polyadenylation assays and pull-down analysis

Cells were resuspended in buffer C (25 mM Tris–HCl pH 7.5, 200 mM NaCl, 10% glycerol, and 0.5 mM TCEP) supplemented with 5 mM imidazole, 0.05% Tween-20, and cOmplete Protease-Inhibitor-Cocktail (Roche). Cells were lysed by sonication, cleared by centrifugation (20 min, $20,000 \times g$, 4°C), and the clarified lysate was purified on Ni-NTA resin (QIAGEN) eluting in buffer C supplemented with 200 mM imidazole. The eluted protein was incubated with Strep-Tactin Sepharose (IBA Lifesciences) beads, washed with 10 column volumes of buffer B, and eluted with buffer C supplemented with 5 mM Desthiobiotin. Strep-Tactin purified mPSF complexes were concentrated in centrifugal filter (Amicon Ultra-15, MWCO 300 kDa, Merck Millipore) to approximately 0.5 mg ml$^{-1}$. To account for impurities, mPSF complex concentrations were assessed on sodium dodecyl sulfate–polyacrylamide gel electrophoresis (SDS–PAGE) and adjusted accordingly (Figure S1C), aliquoted, flash frozen, and stored at −80°C. For use in pull-down analysis, CPSF complexes were used directly after Strep-Tactin purification.

## Purification of mPSF–PAP complex for SEC-MALS analysis

mPSF complexes comprising CPSF160–WDR33$^{1–410}$–CPSF30–hFip1$^{1–198}$ for SEC-MALS analysis were produced as described above with subsequent tag removal by incubation with His$_6$-TEV protease. mPSF (assuming to comprise two hFip1) was supplemented with untagged PAP$^{1–504}$ in 2.5-fold molar excess and 1.2-fold molar excess of a 27-nt RNA substrate based on the SV40 pre-mRNA containing a PAS and a 3′ penta-A tail (CUGC<u>AAUAAA</u>CAACUUAACAACAAAAA). The complex was purified on a Superose 6 column (Cytiva) in 20 mM HEPES pH 8.0, 150 mM KCl, 0.5 mM TCEP. The mPSF–PAP complex was concentrated in centrifugal filter (Amicon Ultra-15, MWCO 100 kDa, Merck Millipore), aliquoted, flash frozen, and stored at −80°C.

## Expression and purification of GST-hFip1 proteins for pull-down analysis

His$_6$-GST-TEV-hFip1$^{1–195}$ and His$_6$-GST-TEV-hFip1$^{1–35}$ were expressed overnight in *E. coli* BL21 star (DE3) cells at 18°C by induction with 0.5 mM IPTG at OD$_{600}$ of about 0.6–0.8. Cells were lysed by sonication in buffer A supplemented with 5 mM imidazole, 1 µM Peptsatin A, and 400 µM AEBSF protease inhibitor. Lysate was cleared by centrifugation (20 min, $20,000 \times g$, 4°C) and clarified lysate was purified on Ni-NTA resin (QIAGEN) eluting in buffer A supplemented with 200 mM imidazole in gravity flow. Eluted protein was loaded on a HiTrap Q FF (Cytiva) anion exchange chromatography column and eluted with a linear gradient from 200 mM to 1 M NaCl over 15 CV in 25 mM Tris–HCl pH 7.5 and 1 mM DTT. Eluting peak fractions were further purified on a Superdex 75 (Cytiva) column equilibrated in 25 mM Tris–HCl pH 7.5, 500 mM NaCl, 1 mM DTT. Protein was concentrated in centrifugal filter (Amicon Ultra-15, MWCO 30 kDa, Merck Millipore), aliquoted, flash frozen, and stored at −80°C. His$_6$-GST-TEV-hFip1$^{36–195}$ was expressed in *E. coli* BL21 star (DE3) cells overnight at 18°C by induction with 0.5 mM IPTG at OD$_{600}$ of about 0.6–0.8. Cells were lysed by sonication in buffer A supplemented with 1 mM DTT, 1 µM Peptsatin A, and 400 µM AEBSF protease inhibitor. Lysate was cleared by centrifugation (20 min, $20,000 \times g$, 4°C). Clarified lysate was subjected to a GSTrap Fast Flow (Cytiva) column and eluted in buffer A supplemented with 1 mM DTT and 10 mM GSH. His$_6$-GST-TEV-hFip1$^{1–35}$ mutants (E22A, E23A; W25A, L26A, Y27A; W25A) were expressed and purified analogously to His$_6$-GST-TEV-hFip1$^{36–195}$ but with buffers containing 500 mM NaCl. All proteins were further purified on a Superdex 200 (Cytiva) column equilibrated in buffer A supplemented with 1 mM DTT. Protein was concentrated in centrifugal filter (Amicon Ultra-15, MWCO 30 kDa, Merck Millipore), aliquoted, flash frozen, and stored at −80°C. His$_6$-GST-TEV-hFip1$^{36–80}$ was expressed in *E. coli* BL21 star (DE3) cells overnight at 18°C by induction with 0.5 mM IPTG at OD$_{600}$ of about 0.6–0.8. Cells were lysed by sonication in buffer A supplemented with 1 mM DTT, 1 µM Peptsatin A, and 400 µM AEBSF protease inhibitor. Lysate was

cleared by centrifugation (20 min, 20,000 × $g$, 4°C). Clarified lysate was subjected to a GSTrap Fast Flow (Cytiva) column, washed with buffer A supplemented with 1 mM DTT prior to elution in buffer A supplemented with 1 mM DTT and 10 mM GSH. Protein was further purified on a Superdex 200 (Cytiva) column equilibrated in buffer A supplemented with 1 mM DTT. Protein was concentrated in centrifugal filter (Amicon Ultra-15, MWCO 30 kDa, Merck Millipore), aliquoted, flash frozen, and stored at −80°C. His$_6$-GST-TEV-hFip1$^{80–195}$ was expressed and purified using the same protocol, but changing to buffer B. His$_6$-GFP-TEV-hFip1$^{1–195}$ mutants (W150E, F161E, and W170E) were expressed and purified following the same purification strategy as for the His$_6$-GFP-TEV-hFip1$^{1–195}$ wt protein.

## Expression and purification of His$_6$-CstF77 proteins for polyadenylation assays

His$_6$-TEV-CstF77$^{21–549}$ wt and mutant (R395A, R402A, K431A) were expressed overnight in *E. coli* BL21 star (DE3) cells at 18°C by induction with 0.5 mM IPTG at OD$_{600}$ of about 0.6–0.8. Cells were lysed by sonication in 40 mM Tris–HCl pH 7.5, 500 mM NaCl, 5 mM imidazole, supplemented with 1 µM Peptsatin A and 400 µM AEBSF protease inhibitor. Lysate was cleared by centrifugation (20 min, 20,000 × $g$, 4°C) and clarified lysate was purified on Ni-NTA resin (QIAGEN) eluting with buffer A supplemented with 250 mM imidazole in gravity flow. Salt concentration and pH of protein sample were reduced to 60 mM NaCl and pH 7.0 by dilution and purified on a HiTrap SP FF (Cytiva) cation exchange chromatography column. Protein was eluted from column with a linear gradient from 60 mM to 1 M NaCl over 10 CV in 20 mM Tris–HCl pH 7.0 and 1 mM DTT. Eluting peak fractions were further purified on a Superdex 200 (Cytiva) column equilibrated in 20 mM Tris–HCl pH 7.5, 200 mM NaCl, 1 mM DTT. Protein was concentrated in centrifugal filter (Amicon Ultra-15, MWCO 100 kDa, Merck Millipore), aliquoted, flash frozen, and stored at −80°C.

## Expression and purification of MBP-CstF77 proteins for pull-down analysis

For use in pull-down analysis, His$_6$-MBP-TEV-CstF77$^{21–549}$ wt and mutant (R395A, R402A, and K431A) were expressed and purified analogous to CstF77 for cocrystallization with hFip1, omitting tag cleavage with His$_6$-TEV protease prior to size-exclusion chromatography.

## Expression and purification of CstF for polyadenylation assay

For expression of CstF complex comprising CstF77, His$_6$-TEV-2xStrepII-TEV-2xStrepII-TEV-CstF64$^{1–198}$, and CstF50 in *Sf9* cells (Thermo Fisher Scientific, cat. no. 11496015; cell line was authenticated and tested for mycoplasma contamination by the manufacturer, no further validation was done by the authors), recombinant baculoviruses were generated according to the Bac-to-Bac Baculovirus expression system (Invitrogen). 2 ml of each P3 virus were used to coinfect 1 l of *Sf9* insect cells at a density of 1.1 × 10$^6$ ml$^{−1}$. Cells were harvested 72 hr postinfection. Cells were resuspended in buffer B supplemented with 10% glycerol, 5 mM imidazole, 0.4% Triton X-100, and cOmplete Protease-Inhibitor-Cocktail (Roche). Cells were lysed by sonication, cleared by centrifugation (20 min, 20,000 × $g$, 4°C) and the clarified lysate was purified on Ni-NTA resin (QIAGEN) eluting in buffer B supplemented with 10% glycerol, 150 mM imidazole, 0.4% Triton X-100. The eluted protein was incubated with Strep-Tactin Sepharose (IBA Lifesciences) beads, washed with 10 column volumes of 25 mM Tris pH 7.5, 120 mM KCl, 10% glycerol, 2 mM MgCl$_2$, 0.5 mM TCEP, 0.05% Tween-20, and eluted with wash buffer supplemented with 5 mM Desthiobiotin. The protein was further purified by size-exclusion chromatography on a Superdex 200 (Cytiva) column eluting with 25 mM Tris pH 7.5, 120 mM KCl, 10% glycerol, 2 mM MgCl$_2$, and 0.5 mM TCEP. Protein was diluted with 2× dilution buffer (25 mM Tris pH 7.5, 10% glycerol, 2 mM MgCl$_2$, 0.5 mM TCEP, 10% Tween-20) to reduce KCl concentration to 50 mM, concentrated in centrifugal filter (Amicon Ultra-15, MWCO 100 kDa, Merck Millipore), aliquoted, flash frozen, and stored at −80°C.

## Expression and purification of CstF complexes for pull-down analysis

CstF complexes comprising CstF77, His$_6$-TEV-2xStrepII-TEV-2xStrepII-TEV-CstF64$^{1–198}$, and CstF50 were expressed as described above. Cells were resuspended in buffer B supplemented with 5 mM imidazole and cOmplete Protease-Inhibitor-Cocktail (Roche). Cells were lysed by sonication, cleared by centrifugation (20 min, 20,000 × $g$, 4°C) and the clarified lysate was purified on Ni-NTA resin (QIAGEN) eluting in buffer B supplemented with 150 mM imidazole. The eluted protein was incubated

with Strep-Tactin Sepharose (IBA Lifesciences) beads, washed with 10 column volumes of buffer B and eluted with buffer B supplemented with 5 mM Desthiobiotin. After overnight cleavage with His$_6$-TEV protease, the protein was further purified by size-exclusion chromatography on a Superdex 200 (Cytiva) column eluting with 20 mM Tris pH 7.5, 150 mM KCl, 1 mM DTT. Protein was concentrated in centrifugal filter (Amicon Ultra-15, MWCO 100 kDa, Merck Millipore), aliquoted, flash frozen, and stored at −80°C.

## Preparation of CPSF30–hFip1 complex for crystallization

Plasmids encoding for His$_6$-TEV-CPSF30$^{118–178}$ and GST-TEV-hFip1$^{130–195}$ was cotransformed and proteins were expressed overnight in *E. coli* BL21 star (DE3) cells at 18°C by addition of IPTG to a final concentration of 0.5 mM at OD$_{600}$ of about 0.6–0.8. Cells were resuspended in buffer A supplemented with 1 µM Pepstatin and 400 µM AEBSF, and lysed by sonication. Lysate was cleared by centrifugation (20 min, 20,000 × *g*, 4°C) and protein was purified on Glutathione Sepharose 4 Fast Flow resin (Cytiva), eluting with buffer A supplemented with 10 mM reduced L-glutathione (GSH). After overnight cleavage with His$_6$-TEV protease, the protein was further purified by size-exclusion chromatography on a Superdex 75 (Cytiva) column with a GSTrap Fast Flow (Cytiva) column in tandem to capture any residual GST tags, eluting with 25 mM Tris–HCl pH 7.5, 150 mM NaCl, and 1 mM DTT. Eluting peak fractions were concentrated in centrifugal filter (Amicon Ultra-15, MWCO 10 kDa, Merck Millipore) to 6.14 mg ml$^{−1}$, flash frozen, and stored at −80°C.

## Preparation of CstF77–hFip1 complex for crystallization

His$_6$-MBP-TEV-CstF77$^{241–549}$ was expressed in *E. coli* BL21 star (DE3) cells at 18°C overnight by addition of IPTG to a final concentration of 0.5 mM at OD$_{600}$ of about 0.6–0.8. Cells were resuspended in buffer containing buffer A supplemented with 1 mM DTT, 1 µM Pepstatin, and 400 µM AEBSF, and lysed by sonication. Lysate was cleared by centrifugation (20 min, 20,000 × *g*, 4°C) and protein was purified on amylose resin (New England Biolabs) including a high salt wash with buffer containing 25 mM Tris–HCl pH 7.5, 500 mM NaCl, and 1 mM DTT prior to elution with buffer A supplemented with 1 mM DTT and 10 mM maltose. After digestion with His$_6$-TEV protease, the tags and protease were removed from the protein by passage through a Ni-NTA superflow cartridge (QIAGEN). The protein was further purified by size-exclusion chromatography on a Superdex 200 Increase (Cytiva) column, eluting with 20 mM HEPES pH 7.5, 150 mM KCl, 1 mM TCEP. His$_6$-GST-TEV-hFip1$^{1–35}$ was expressed in *E. coli* BL21-AI (Invitrogen) cells overnight at 18°C by induction with 0.2% arabinose at OD$_{600}$ of 0.8. Cells were lysed by high-pressure cell disruption at 25 kpsi in buffer A supplemented with 1 mM DTT, 1 µM Pepstatin, and 400 µM AEBSF protease inhibitor. Lysate was cleared by centrifugation (20 min, 20,000 × *g*, 4°C) and clarified lysate was subjected to a GSTrap Fast Flow (Cytiva) column, washed with 25 mM Tris–HCl pH 7.5, 500 mM NaCl, and 1 mM DTT prior to elution in buffer A supplemented with 1 mM DTT and 10 mM GSH. Affinity tag was cleaved from protein using His$_6$-MBP-TEV protease while dialyzing into buffer A supplemented with 1 mM DTT and hFip1$^{1–35}$ was further purified by size-exclusion chromatography on a Superdex 75 (Cytiva) column into 25 mM HEPES pH 7.5, 150 mM KCl, 1 mM DTT. The absolute mass of hFip1$^{1–35}$ (4.1 kDa) was confirmed with ESI-MS analysis. Peak fractions of both CstF77 and hFip1 were pooled individually, concentrated, flash frozen, and stored at −80°C.

## CPSF30–hFip1 complex crystallization and structure determination

The CPSF30–hFip1 complex was crystallized at 20°C using the hanging drop vapour diffusion method by mixing 0.5 µl of protein at 6.14 mg ml$^{−1}$ with 0.5 µl of reservoir solution containing either 1.8 M (NH$_4$)SO$_4$, 0.1 M Bis-Tris pH 6.5 (native dataset) or 1.626 M (NH$_4$)SO$_4$, 0.1 M Bis-Tris pH 6.5 (zinc SAD dataset). Crystals were transferred into reservoir solution supplemented with 20% (vol/vol) glycerol for cryo-protection prior to flash-cooling by plunging into liquid nitrogen. X-ray diffraction data were recorded at beam line X06DA (PXIII) at Swiss Light Source (Paul Scherrer Institute, Villigen, Switzerland) on a PILATUS 2 M-F (Dectris) detector, at a wavelength of 1.28095 Å using an oscillation range of 0.1° and an exposure time of 0.1 s per image while rotating the crystal through 360°. Detailed data collection statistics are listed in *Table 1*. Diffraction data were processed with XDS (*Kabsch, 2010*) in space group P2$_1$, with four complex copies in the asymmetric unit and the presence of pseudmerohedral twinning. Twin law h, -k, -h-l was determined using phenix.xtriage (*Zwart and Grosse-Kunstleve, 2005*) comprising a twin fraction of approximately 48%. Exploiting the presence of zinc ions bound

**Table 1.** Crystallographic data collection and refinement statistics.

| | hFip1–CPSF30 | hFip1–CstF77 |
|---|---|---|
| **Data collection** | | |
| Space group | $P2_1$ | $P6_122$ |
| Cell dimensions | | |
| $a, b, c$ (Å) | 60.127, 115.125, 66.444 | 157.612, 157.612, 161.005 |
| $\alpha, \beta, \gamma$ (°) | 90, 116.781, 90 | 90, 90, 120 |
| Wavelength (Å) | 1.2809 | 1.0000 |
| Resolution (Å) | 48.65–2.201 (2.28–2.201) | 56.31–2.55 (2.641–2.55) |
| Total reflections | 226,720 (15,294) | 1,577,004 (162,259) |
| Unique reflections | 37,698 (3244) | 38,981 (3836) |
| $R_{merge}$ (%) | 7.5 (95.9) | 9.2 (186.1) |
| $R_{pim}$ (%) | 3.2 (46.9) | 1.5 (28.8) |
| $I/\sigma I$ | 13.5 (1.1) | 36.0 (2.6) |
| $Cc(1/2)$ | 0.998 (0.557) | 1 (0.836) |
| Completeness (%) | 92.3 (80.22) | 99.96 (100.00) |
| Redundancy | 6.0 (4.7) | 40.5 (42.3) |
| | | |
| **Refinement** | | |
| Resolution (Å) | 48.65–2.201 | 56.31–2.55 |
| No. reflections | 37,698 | 38,975 |
| $R_{work}$ / $R_{free}$ | 0.2406/0.2622 | 0.2410/0.2647 |
| No. non-hydrogen atoms | | |
| Protein | 4607 | 5188 |
| Ligand/ion | 8 | 98 |
| Water | 67 | 25 |
| $B$-factors (Å$^2$) | | |
| Protein | 56.53 | 65.34 |
| Ligand/ion | 63.69 | 69.46 |
| Water | 49.83 | 55.9 |
| R.m.s. deviations | | |
| Bond lengths (Å) | 0.008 | 0.009 |
| Bond angles (°) | 1.03 | 1.1 |
| Ramachandran plot | | |
| % favored | 95.83 | 97.9 |
| % allowed | 4.17 | 2.1 |
| % outliers | 0 | 0 |

Values in parentheses are for highest resolution shell.

to CPSF30, phase determination was performed by single-wavelength anomalous diffraction (SAD) followed by phasing and density modification with autoSHARP (*Vonrhein et al., 2007*). A homology model based on CPSF30 ZF2 (PDB ID: 6FUW) was fitted into the electron density in Coot (*Emsley and Cowtan, 2004*), followed by automated model building using phenix.autobuild (*Terwilliger et al., 2008*). The structure was completed by iterative cycles of manual model building in Coot and refinement with phenix.refine (*Adams et al., 2010*). Molecular models were visualized using PyMOL (*Schrödinger LLC, 2021*).

## CstF77–hFip1 complex structure determination

A 1.5-fold molar excess of hFip1 was added per CstF77 molecule (corresponding to a threefold molar excess to a dimer of CstF77) and concentrated to 13.7 mg ml$^{-1}$ ($A_{280}$ = 23.36) using a centrifugal filter (Amicon Ultra-0.5, MWCO 3 kDa, Merck Millipore) prior to crystallization. The CstF77–hFip1 complex was crystallized using the sitting drop vapor diffusion method by mixing 0.1 µl protein with 0.1 µl reservoir solution containing 0.1 M Bicine pH 9.0, 10% (wt/vol) PEG 20 k, 2% (vol/vol) Dioxane. Crystals were cryo-protected by transfer into reservoir solution supplemented with 24% (vol/vol) glycerol prior to flash-cooling with liquid nitrogen. X-ray diffraction data were recorded at beam line X06SA (PXI) at Swiss Light Source (Paul Scherrer Institute, Villigen, Switzerland) on an EIGER 16 M (Dectris) detector, using an oscillation range of 0.2° and an exposure time of 0.1 s per image while rotating the crystal through 360°. Detailed data collection statistics are listed in *Table 1*. Diffraction data were processed with Autoproc (*Vonrhein et al., 2011*) in space group *P*6$_1$22. The structure was solved using residues 241–549 of murine CstF77 (PDB ID: 2OOE) as search model for phasing with molecular replacement (MR) in phenix.phaser (*McCoy et al., 2007*). A total of two CstF77 molecules could be placed into the electron density, corresponding to a dimer. After rigid-body refinement of the molecular replacement solution, the structure was completed by iterative cycles of manual model building in Coot, including the placement of the hFip1 peptides into the electron density unoccupied by CstF77, and refinement with phenix.refine (*Adams et al., 2010*). Molecular models were visualized using PyMOL (*Schrödinger LLC, 2021*).

## Pull-down assays

For all pull-down assays, bound proteins were eluted with 1× SDS–PAGE loading buffer on ice and analyzed by SDS–PAGE on 4–20% Mini-PROTEAN TGX Precast Protein Gels (Bio-Rad) without prior heating to preserve the GFP fluorescence. GFP fluorescence was visualized on a Typhoon FLA 9500 laser scanner (Cytiva) at 473 nm and subsequently stained with Coomassie brilliant blue R250.

### Pull-down analysis of mPSF–PAP interaction

Strep-Tactin purified mPSF complexes were incubated with 30 µl Anti-FLAG M2 magnetic beads (Sigma-Aldrich) equilibrated in FLAG wash buffer (25 mM Tris–HCl pH 8.0, 200 mM NaCl, 0.1% Tween-20) and gently agitated at 4°C for 1 hr. The beads were washed three times with 0.5 ml of FLAG wash buffer supplemented with 2 mM MgCl$_2$ and 165 µg PAP and 17 µg 27-nt substrate based on the SV40 pre-mRNA containing a PAS and a 3′ penta-A tail (CUGC<u>AAUAAA</u>CAACUUAACAACAAAAA) were added to the mixture and gently agitated at 4°C for 1 hr. The beads were washed three times with 0.5 ml of FLAG wash buffer supplemented with 2 mM MgCl$_2$ and the bound protein was eluted with 1× SDS–PAGE loading buffer supplemented with 100 µg ml$^{-1}$ 3× FLAG peptide (Sigma-Aldrich) on ice. FLAG elutions were analyzed by SDS–PAGE on 4–20% Mini-PROTEAN TGX Precast Protein Gels (Bio-Rad) without prior heating to preserve the GFP fluorescence. GFP fluorescence was visualized on a Typhoon FLA 9500 laser scanner (Cytiva) at 473 nm and subsequently stained with Coomassie brilliant blue R250. For equal mPSF complex concentrations to compare the corresponding GFP-hFip1 and GFP-PAP fluorescence intensities, loading volumes were adjusted according to CPSF160 band intensities. Beads control loading volume corresponds to the maximum mPSF sample loading volume.

### Pull-down analysis of hFip1–CstF77 interaction

For pull-down analysis with purified hFip1 and CstF77 proteins (wt and mutants), 10 µg of purified His$_6$-GST-hFip1 protein was immobilized on 15 µl Glutathione Sepharose 4 Fast Flow beads (Cytiva) and washed three times with 0.5 ml pull-down wash buffer (20 mM Tris–HCl pH 7.5, 200 mM NaCl, 0.05% Tween-20, 0.5 mM TCEP). His$_6$-MBP-CstF77 protein was added to the immobilized protein at

fourfold molar excess and incubated gently agitating at 4°C for 1 hr followed by washing three times with 0.5 ml of pull-down wash buffer. The bound protein was eluted at room temperature by adding 1× SDS–PAGE loading buffer and analyzed by SDS–PAGE on 4–20% Mini-PROTEAN TGX Precast Protein Gels (Bio-Rad) stained with Coomassie brilliant blue R250.

## Pull-down analysis of hFip1–CstF77–PAP interaction

For competitive pull-down analysis of both CstF77 and PAP with hFip1$^{1–195}$, 5 µg of purified His$_6$-GST-hFip1$^{1–195}$ protein was immobilized on 15 µl Glutathione Sepharose 4 Fast Flow beads (Cytiva) equilibrated in pull-down wash buffer, gently agitated at 4°C for 1 hr, and washed three times with 0.5 ml pull-down wash buffer. His$_6$-MBP-CstF77 and His$_6$-GFP-PAP were incubated with the bait, either individually or combined (1:1) at fourfold molar excess, as well as adding a 32-fold molar excess of one protein while keeping the other at fourfold molar excess, resulting in an eightfold excess of one protein over the other (8:1, 1:8).

## Pull-down analysis of hFip1–CPSF30 interaction

For pull-down analysis of hFip1 mutants binding to CPSF30, 120 µg of purified His$_6$-MBP-TEV-CPSF30$^{1–243}$ wt protein was incubated with 120 µl amylose resin (NEB) equilibrated in pull-down wash buffer and gently agitated at 4°C for 1 hr. The beads were washed three times with 0.5 ml of pull-down wash buffer and equally distributed in four tubes. His$_6$-GFP-TEV-hFip1$^{1–195}$ wt and point mutants (W150E, F161E, and W170E) were added in fivefold molar excess to the beads. After incubation at 4°C for 1 hr, gently agitated, unbound protein was washed off by adding three times 0.5 ml pull-down wash buffer. For pull-down analysis of the hFip1 interaction with the ZF of CPSF30, 15 µg of His$_6$-MBP-TEV-CPSF30$^{1–243}$ wt and ZF mutants were incubated each with 30 µl amylose resin (NEB) equilibrated in pull-down wash buffer and gently agitated at 4°C for 1 hr. Unbound protein was washed off three times with 0.5 ml of pull-down wash buffer and His$_6$-GFP-TEV-hFip1$^{1–195}$ wt was added in fourfold molar excess to the resin and incubated at 4°C for 1 hr, gently agitated. Beads were washed three times with 0.5 ml pull-down wash buffer.

## Pull-down analysis of hFip1–PAP interaction

For pull-down analysis of the hFip1:PAP interaction, 20 µg of His$_6$-GST-TEV-hFip1 truncation constructs (hFip1$^{1–195}$, hFip1$^{36–195}$, hFip1$^{80–195}$, hFip1$^{36–80}$, and hFip1$^{1–35}$) were incubated each with 15 µl Glutathione Sepharose 4 Fast Flow beads (Cytiva) equilibrated in pull-down wash buffer and gently agitated at 4°C for 1 hr. Unbound protein was washed off three times with 0.5 ml of pull-down wash buffer and His$_6$-GFP-TEV-PAP$^{1–504}$ was added in fourfold molar excess to the resin and incubated at 4°C for 1 hr, gently agitated. Beads were washed three times with 0.5 ml pull-down wash buffer.

## Pull-down analysis of mPSF–CstF77 interaction

For pull-down analysis of the mPSF:CstF77 interaction, Ni-IMAC purified mPSF complexes from *Sf9* cells containing hFip1$^{1–195}$ and N-terminal truncations hereof (hFip1$^{36–195}$, hFip1$^{80–195}$) were incubated with 20 µl Strep-Tactin (IBA Lifesciences) beads in buffer containing 20 mM HEPES–KOH pH 8.0, 150 mM KCl, 0.05% Tween-20, 0.5 mM TCEP and gently agitated at 4°C for 1 hr. Unbound protein was washed off three times with 0.5 ml of buffer containing 20 mM HEPES–KOH pH 8.0, 150 mM KCl, 0.05% Tween-20, 0.5 mM TCEP, and 10 µg His$_6$-MBP-TEV-CstF77$^{21–549}$ was added to the resin and incubated at 4°C for 1 hr, gently agitated. Beads were again washed three times with 0.5 ml.

## Pull-down analysis of hFip1–CstF interaction

For pull-down analysis with purified hFip1, CstF complex, and CstF77, 10 µg of purified His$_6$-GST-hFip1 protein was immobilized on 15 µl Glutathione Sepharose 4 Fast Flow beads (Cytiva) and washed three times with 0.5 ml pull-down wash buffer (20 mM Tris–HCl pH 7.5, 200 mM NaCl, 0.05% Tween-20, 0.5 mM TCEP). CstF complex and His$_6$-MBP-CstF77 protein were added to the immobilized protein at fourfold molar excess and incubated gently agitating at 4°C for 1 hr followed by washing three times with 0.5 ml of pull-down wash buffer. The bound protein was eluted at room temperature by adding 1× SDS–PAGE loading buffer and analyzed by SDS–PAGE on 4–20% Mini-PROTEAN TGX Precast Protein Gels (Bio-Rad) stained with Coomassie brilliant blue R250.

## In vitro polyadenylation assays

Reaction conditions for pre-mRNA polyadenylation were adjusted for the individual need of each assay and evolved over the course of the project. To account for potential impurities and to ensure equal mPSF complex concentrations, CPSF160 band intensities were assessed on SDS–PAGE (Figure S1C) and concentrations adjusted accordingly. All polyadenylation reactions were performed in polyadenylation buffer (25 mM Tris–HCl pH 7.5, 10% glycerol, 50 mM KCl, 2 mM MgCl$_2$, 0.05% Tween-20, 1 mM DTT) with 20 nM 5′ Cy5-labeled a 27-nt RNA substrate based on the SV40 pre-mRNA containing a PAS and a 3′ penta-A tail (CUGC<u>AAUAAA</u>CAACUUAACAACAAAAA). All proteins were first diluted in polyadenylation buffer. Protein–RNA mixes with a total volume of 36 µl were prepared on ice, preheated at 37°C for 1 min and reaction was started by the addition of preheated 12 µl ATP at 37°C. Reaction mix for polyadenylation assay with CPSF30 ZF4/ZF5 mutants (*Figure 2A*) contained 80 nM mPSF complexes, 1.46 µM PAP, and a final concentration of 4 µM ATP. Reaction mix for polyadenylation assay with hFip1 truncations (*Figure 2C*) contained 40 nM mPSF complexes, 120 nM PAP, and a final concentration of 500 µM ATP. Reaction mix for polyadenylation assay with CstF77 (*Figure 4B*) at 80 (denoted 0.5), 160 (denoted +), 320, 640, or 1280 nM (denoted 2, 4, and 8, respectively) contained 80 nM mPSF complexes, 160 nM PAP, and a final concentration of 500 µM ATP. Reaction mix for polyadenylation assay with holo-CstF at 80 (denoted +), 160, 320, and 640 nM (denoted 2, 4, and 8, respectively) contained 80 nM mPSF complexes, 80 nM PAP, and a final concentration of 500 µM ATP. Time points were taken at indicated times (1 and 10 min) and polyadenylation stopped by the addition of Ethylenediaminetetraacetic acid (EDTA) with final concentration of 166 mM and incubation with 20 µg Proteinase K at 37°C for 10 min. The reactions were mixed with 2× denaturing PAGE loading dye (90% formamide, 5% glycerol, 25 mM EDTA, bromophenol blue), incubated at 95°C for 10 min and analyzed on a 15% denaturing PAGE gel containing 8 M urea and 0.5× Tris–borate–EDTA (TBE) buffer with low range ssRNA ladder (NEB). Gel was subsequently stained with SYBR Gold nucleic acid stain (Invitrogen). In-gel fluorescence of 5′ Cy5-labeled RNA and SYBR Gold-stained ssRNA ladder was visualized with Typhoon FLA 9500 laser scanner (Cytiva) at 635 and 473 nm, respectively.

### Polyadenylation assay with titration of CstF77

Polyadenylation reactions were performed according to standard procedure described above. Reaction mix for polyadenylation assay with varying CstF77 or CstF77[mut] concentrations (*Figure 4—figure supplement 1A*) contained 80 nM of mPSF complex (CPSF160-WDR33[1–410]-CPSF30-hFip1[1–198]), 160 nM PAP[1–504], and a final concentration of 500 µM ATP. His$_6$-TEV-CstF77[21–549] or His$_6$-TEV-CstF77[mut,21–549] were added at 160 (denoted as +) or 1280 nM (denoted 8).

### Polyadenylation assay with L3 mRNA and titration of CstF77

Polyadenylation reactions were performed according to the standard procedure described above with 20 nM 5′ Cy5-labeled 38-nucleotide L3 mRNA-based substrate. Reaction mix for polyadenylation assay with varying CstF77 concentrations (*Figure 4—figure supplement 1B*) contained 80 nM of mPSF complex (CPSF160–WDR33[1–410]–CPSF30–hFip1[1–198]), 160 nM PAP[1–504], and a final concentration of 500 µM ATP. His$_6$-TEV-CstF77[21–549] was added at 160 (denoted as +) or 1280 nM (denoted 8).

## SEC-MALS analysis

SEC-MALS was carried out on an HPLC system (Agilent LC1100, Agilent Technologies) coupled to an Optilab rEX refractometer and a miniDAWN three-angle light-scattering detector (Wyatt Technology). Data analysis was performed using the ASTRA software (version 7.3.2; Wyatt Technology).

### SEC-MALS analysis of hFip1–CPSF30 complex

For unambigous determination of the stoichiometry of the respective hFip1–CPSF30 complexes, tagged proteins were used to increase the molecular weight difference between the 2:1 and 1:1 complexes of hFip1–CPSF30. Stoichiometry of the complexes was determined injecting 33 µg His$_6$-MBP-CPSF30[1–243] (wt and mutants) and fourfold molar excess of His$_6$-GFP-hFip1[1–195] premixed in a total injection volume of 100 µl. Proteins were separated on a Superdex 200 10/300 GL column (Cytiva) run at 0.5 ml/min at room temperature in 20 mM Tris–HCl pH 7.5, 200 mM NaCl, 0.5 mM TCEP (pH was adjusted at room temperature).

## SEC-MALS analysis of mPSF–PAP complex

Stoichiometry of the mPSF–PAP complex was determined injecting 50 µg of prepurified mPSF-PAP comprising CPSF160–WDR33[1–410]–CPSF30–2xhFip1[1–198], PAP[1–504], and 27-nt SV40 PAS-containing mRNA in a total injection volume of 100 µl. In a second run, prepurified 50 µg of mPSF:PAP was spiked with additional 41.6 µg PAP[1–504] (fivefold molar excess) in a total injection volume of 100 µl to test whether excess PAP can lead to a stable 1:2 complex of mPSF and PAP. Proteins were separated on a Superose 6 10/300 GL column (Cytiva) run at 0.5 ml/min at room temperature in 20 mM HEPES pH 8.0, 150 mM KCl, 0.5 mM TCEP (pH was adjusted at room temperature).

## Multiple sequence alignment

The multiple sequence alignment of hFip1 orthologs was produced with MAFFT version 7 (*Katoh et al., 2018*) and visualized using Jalview (*Waterhouse et al., 2009*). Input sequences are listed in *Supplementary file 3*.

## Analysis of interaction interfaces

Buried surface area of the hFip1–CPSF30 interaction interface was calculated using the PDBe PISA (Proteins, Interfaces, Structures and Assemblies) tool (*Krissinel and Henrick, 2007*).

## 3D density map analysis

Visualization and analysis of the 3D density map for CPSF160–WDR33–CPSF30–PAS RNA–CstF77 complex (EMD-20861) were performed with UCSF Chimera (*Pettersen et al., 2004*), developed by the Resource for Biocomputing, Visualization, and Informatics at the University of California, San Francisco. The 3D density map was segmented and color coded based on the corresponding atomic model (PDB ID: 6URO). The CstF77–hFip1 crystal structure from this study was superimposed onto the atomic model of CstF77.

## Bioinformatic analysis of CstF77

Color-coded electrostatic surface representation of CstF77 was generated for the biological assembly of murine CstF77[20–549] (PDB ID: 2OOE) in PyMOL 2.5.0 (*Schrödinger LLC, 2021*) using the protein contact potential option. The CstF77[241–549]–hFip1[1–35] structure from this study and cryo-EM structure of human CPSF160–WDR33–CPSF30–PAS RNA–CstF77 complex (PDB ID: 6URO) were superimposed onto murine CstF77 using PyMOL's align command to identify and visualize the hFip1- and mPSF-binding regions, respectively. Analysis of evolutionary conservation of CstF77 was carried out using the ConSurf web server (*Ashkenazy et al., 2016*) with murine CstF77[20–549] (PDB ID: 2OOE) as input and applying standard settings (sequence alignment with MAFFT, homologs taken from UniRef90). The degree of conservation was visualized in PyMOL by color coding (green: variable, violet: conserved) the protein surface according to the conservation scores which are written into the tempFactor column of the ConSurf web server output PDB file.

## Acknowledgements

We thank Birgit Dreyer for assistance with SEC-MALS, Beat Blattmann (University of Zurich Protein Crystallization Center) for performing crystallization screens, and Levi Kopp for assistance with protein crystallization. We thank Vincent Olieric, Takashi Tomizaki, and Meitian Wang (Swiss Light Source, Paul Scherrer Institute) for assistance with crystallographic data collection. We are grateful to Stefanie Jonas and members of the Jinek laboratory for critical reading of the manuscript. This work was supported by Boehringer Ingelheim Fonds PhD Fellowship and by the National Center for Competence in Research (NCCR) RNA & Disease, funded by the Swiss National Science Foundation.

## Additional information

### Funding

| Funder | Grant reference number | Author |
|---|---|---|
| Schweizerischer Nationalfonds zur Förderung der Wissenschaftlichen Forschung | NCCR RNA and Disease | Martin Jinek |
| Boehringer Ingelheim Fonds | | Lena Maria Muckenfuss |
| Howard Hughes Medical Institute | 55008735 | Martin Jinek |

The funders had no role in study design, data collection, and interpretation, or the decision to submit the work for publication.

### Author contributions

Lena Maria Muckenfuss, Conceptualization, Formal analysis, Investigation, Methodology, Writing - original draft; Anabel Carmen Migenda Herranz, Franziska Maria Boneberg, Investigation, Methodology; Marcello Clerici, Conceptualization, Investigation; Martin Jinek, Conceptualization, Formal analysis, Supervision, Funding acquisition, Investigation, Methodology, Writing - original draft, Project administration, Writing - review and editing

### Author ORCIDs

Lena Maria Muckenfuss ⓘ http://orcid.org/0000-0003-3558-7211
Martin Jinek ⓘ http://orcid.org/0000-0002-7601-210X

### Decision letter and Author response

Decision letter https://doi.org/10.7554/eLife.80332.sa1
Author response https://doi.org/10.7554/eLife.80332.sa2

## Additional files

### Supplementary files

• Supplementary file 1. List of protein expression constructs used in this study.
• Supplementary file 2. Sequences of RNA oligonucleotides used in this study.
• Supplementary file 3. hFip1 ortholog input sequences used for multiple sequence alignment.
• MDAR checklist

### Data availability

X-ray diffraction data (atomic coordinates and structure factors) have been submitted to the PDB and will be released upon publication.

The following datasets were generated:

| Author(s) | Year | Dataset title | Dataset URL | Database and Identifier |
|---|---|---|---|---|
| Muckenfuss L, Jinek M | 2022 | Crystal structure of human CPSF30 in complex with hFip1 | https://www.rcsb.org/structure/7ZYH | RCSB Protein Data Bank, 7ZYH |
| Muckenfuss L, Jinek M | 2022 | Crystal structure of human CstF77 in complex with hFip1 | https://www.rcsb.org/structure/7ZY4 | RCSB Protein Data Bank, 7ZY4 |

The following previously published datasets were used:

| Author(s) | Year | Dataset title | Dataset URL | Database and Identifier |
|---|---|---|---|---|
| Sun Y, Zhang Y, Walz T, Tong L | 2019 | Cryo-EM structure of human CPSF160-WDR33-CPSF30-PAS RNA-CstF77 complex | https://www.rcsb.org/structure/6URO | RCSB Protein Data Bank, 6URO |
| Sun Y, Zhang Y, Walz T, Tong L | 2019 | Cryo-EM structure of human CPSF160-WDR33-CPSF30-PAS RNA-CstF77 complex | https://www.ebi.ac.uk/emdb/EMD-20861 | Electron Microscopy Data Bank, EMD-20861 |

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

# Appendix 1

## Appendix 1—key resources table

| Reagent type (species) or resource | Designation | Source or reference | Identifiers | Additional information |
|---|---|---|---|---|
| Strain, strain background (*Escherichia coli*) | BL21 star (DE3) | Thermo Fisher scientific | BL21 star (DE3) | Chemically competent cells |
| Strain, strain background (*Escherichia coli*) | BL21(DE3)-AI | Thermo Fisher scientific | BL21(DE3)-AI | Chemically competent cells |
| Strain, strain background (*Escherichia coli*) | Rosetta2 (DE3) | Novagen | Rosetta2 (DE3) | Chemically competent cells |
| Cell line (*Spodoptera frugiperda*) | Sf9 | Thermo Fisher Scientific | Cat. #11496015 | |
| Recombinant DNA reagent | pLM B042; pLM B043 (plasmid) | This paper | | Holo-CstF |
| Recombinant DNA reagent | pLM B092 (plasmid) | This paper | | MBP-CstF77 |
| Recombinant DNA reagent | pLM B123 (plasmid) | This paper | | CstF77 |
| Recombinant DNA reagent | pLM B142 (plasmid) | This paper | | PAP |
| Recombinant DNA reagent | pLM B156 (plasmid) | This paper | | MBP-PAP |
| Recombinant DNA reagent | pLM B157 (plasmid) | This paper | | GFP-PAP |
| Recombinant DNA reagent | pLM B164 (plasmid) | This paper | | MBP-CstF77$^{21-549}$ or MBP-CstF77 |
| Recombinant DNA reagent | pLM B168 (plasmid) | This paper | | MBP-CstF77$_{mut}$ |
| Recombinant DNA reagent | pLM B170 (plasmid) | This paper | | CstF77$^{mut}$ |
| Recombinant DNA reagent | pMC B051 (plasmid) | This paper | | CPSF30$^{ZF4-ZF5}$ |
| Recombinant DNA reagent | pMC B054 (plasmid) | This paper | | MBP-CPSF30$^{ZF4-ZF5}$ |
| Recombinant DNA reagent | pMC B055 (plasmid) | This paper | | CPSF30 ZF4 mutant |
| Recombinant DNA reagent | pMC B056 (plasmid) | This paper | | CPSF30 ZF5 mutant |
| Recombinant DNA reagent | pMC B057 (plasmid) | This paper | | CPSF30 ZF4 and ZF5 mutant |
| Recombinant DNA reagent | pMC B058 (plasmid) | This paper | | CPSF30 ZF4 mutant |
| Recombinant DNA reagent | pMC B059 (plasmid) | This paper | | CPSF30 ZF4 mutant |
| Recombinant DNA reagent | pMC B060 (plasmid) | This paper | | CPSF30 ZF5 mutant |
| Recombinant DNA reagent | pMC B061 (plasmid) | This paper | | CPSF30 ZF5 mutant |
| Recombinant DNA reagent | pMC B062 (plasmid) | This paper | | CPSF30 ZF4 and ZF5 mutant |
| Recombinant DNA reagent | pMC B063 (plasmid) | This paper | | CPSF30 ZF4 and ZF5 mutant |
| Recombinant DNA reagent | pMC C011 (plasmid) | This paper | | hFip1$^{CD}$ |
| Recombinant DNA reagent | pMC C015 (plasmid) | This paper | | GST-hFip1 fragment or hFip1$^{80-195}$ |

*Appendix 1 Continued on next page*

*Appendix 1 Continued*

| Reagent type (species) or resource | Designation | Source or reference | Identifiers | Additional information |
|---|---|---|---|---|
| Recombinant DNA reagent | pMC C030 (plasmid) | This paper | | hFip1$^{CD}$ |
| Recombinant DNA reagent | pMC C049 (plasmid) | This paper | | GFP-hFip1 |
| Recombinant DNA reagent | pMC C050 (plasmid) | This paper | | GST-hFip1 fragment or hFip1$^{36-80}$ |
| Recombinant DNA reagent | pMC C059 (plasmid) | This paper | | GST-hFip1 fragment, GST-hFip1$^{1-35}$, or hFip1$^{1-35}$ |
| Recombinant DNA reagent | pMC C060 (plasmid) | This paper | | GST-hFip1 fragment, GST-hFip1$^{1-195}$, or hFip1$^{1-195}$ |
| Recombinant DNA reagent | pMC C066 (plasmid) | This paper | | His$_6$-GFP-TEV-hFip1$^{1-195}$ point mutant |
| Recombinant DNA reagent | pMC C067 (plasmid) | This paper | | His$_6$-GFP-TEV-hFip1$^{1-195}$ point mutant |
| Recombinant DNA reagent | pMC C068 (plasmid) | This paper | | His$_6$-GFP-TEV-hFip1$^{1-195}$ point mutant |
| Recombinant DNA reagent | pMC C073 (plasmid) | This paper | | GST-hFip1 fragment, GST-hFip1$^{36-195}$, or hFip1$^{36-195}$ |
| Recombinant DNA reagent | pMC C093 (plasmid) | This paper | | mutant GST-hFip1$^{1-35}$ |
| Recombinant DNA reagent | pMC C094 (plasmid) | This paper | | mutant GST-hFip1$^{1-35}$ |
| Recombinant DNA reagent | pMC C096 (plasmid) | This paper | | mutant GST-hFip1$^{1-35}$ |
| Recombinant DNA reagent | pMC N015 (plasmid) | This paper | | mPSF |
| Recombinant DNA reagent | pMC N018 (plasmid) | This paper | | mPSF |
| Recombinant DNA reagent | pMC N018A (plasmid) | This paper | | mPSF |
| Recombinant DNA reagent | pMC N018C-2 (plasmid) | This paper | | mPSF |
| Recombinant DNA reagent | pMC N018G (plasmid) | This paper | | mPSF |
| Recombinant DNA reagent | pMC N018G-0 (plasmid) | This paper | | mPSF |
| Recombinant DNA reagent | pMC N018G-8 (plasmid) | This paper | | mPSF |
| Recombinant DNA reagent | pMC N018G-10 (plasmid) | This paper | | mPSF |
| Recombinant DNA reagent | pMC N018G-12 (plasmid) | This paper | | mPSF |
| Recombinant DNA reagent | pMC N018G-14 (plasmid) | This paper | | mPSF |
| Recombinant DNA reagent | pMC N018G-15 (plasmid) | This paper | | mPSF |
| Recombinant DNA reagent | pMC N018G-21 (plasmid) | This paper | | FLAG-epitope-tagged mPSF |
| Recombinant DNA reagent | pMC N018G-22 (plasmid) | This paper | | FLAG-epitope-tagged mPSF |
| Recombinant DNA reagent | pMC N018G-23 (plasmid) | This paper | | FLAG-epitope-tagged mPSF |
| Recombinant DNA reagent | pMC N018G-24 (plasmid) | This paper | | FLAG-epitope-tagged mPSF |
| Recombinant DNA reagent | pMC N018H (plasmid) | This paper | | mPSF |

*Appendix 1 Continued*

| Reagent type (species) or resource | Designation | Source or reference | Identifiers | Additional information |
|---|---|---|---|---|
| Recombinant DNA reagent | pMC N018I (plasmid) | This paper | | mPSF |
| Recombinant DNA reagent | pMC N018J (plasmid) | This paper | | mPSF |
| Recombinant DNA reagent | pMC N018K (plasmid) | This paper | | mPSF |
| Sequence-based reagent | rLM 011 | This paper | | 27 nt RNA substrate based on SV40 pre-mRNA; CUGCAAUAAACAACUU AACAACAAAAA |
| Sequence-based reagent | rLM 015 | This paper | | 5' Cy5-labeled 27 nt RNA substrate based on SV40 pre-mRNA; CUGC AAUAAACAACUUAACGUCAAAAA |
| Sequence-based reagent | rLM 016 | This paper | | 5' Cy5-labeled 27 nt RNA substrate based on SV40 pre-mRNA; CUGC AGUACACAACUUAACGUCAAAAA |
| Sequence-based reagent | rLM 031 | This paper | | 5' Cy5-labeled 38 nt RNA substrate based on adenoviral L3 pre-mRNA; ACUUUCAAUAAAGGCAAAUGUUUUUAUUUGUACAAAAA |

