## [Editor Report]

This study explores the structural and biochemical basis for Fip1 interactions within the cleavage and polyadenylation machinery – notably with CPSF30 and CstF77. Overall, the significance of the study is that it provides valuable mechanistic insight into the function of Fip1 in the cleavage and polyadenylation machinery. The data presented in the paper are compelling and the authors use a combination of structural biology and biochemistry to present their case. This study will be of interest to those focusing on mRNA biosynthesis and the biophysical properties of RNA binding proteins.

---

## [Decision Letter]

**Decision letter after peer review:**

Thank you for submitting your article "Fip1 is a multivalent interaction scaffold for processing factors in human mRNA 3' end biogenesis" for consideration by *eLife*. Your article has been reviewed by 3 peer reviewers, one of whom is a member of our Board of Reviewing Editors, and the evaluation has been overseen by James Manley as the Senior Editor. The following individual involved in the review of your submission has agreed to reveal their identity: William Marzluff (Reviewer #3).

The reviewers have discussed their reviews with one another, and the Reviewing Editor has drafted this to help you prepare a revised submission. We note that no further experimentation is requested. Rather, textual changes are suggested to further strengthen the manuscript. The two critical components of these suggested changes are highlighted in the 'essential revisions' but we encourage the authors to consider response/changes in accordance with other points raised by the Reviewers as described below.

Essential revisions:

1) Through thoughtful discussion, please address differences in approaches used to support the idea that a single copy of PAP exists in the complex in contrast to a previous publication. There are caveats to both studies and distinct experimental differences that should be highlighted to temper their overall narrative.

2) Reviewers were particularly interested in the CstF77 data, specifically the interpretation. This is potentially important data to the field and should be discussed more in this study.

*Reviewer #1 (Recommendations for the authors):*

In the study by Muckenfuss et al., the authors provide two independent crystal structures of Fip1 in complex with CPSF30 or CstF77 as a basis for biochemical experiments exploring the function of Fip1 in polyadenylation. The structure of CPSF30/Fip1 is validated using polyadenylation assays and the authors conclude that while two molecules of Fip1 interact with a single molecule of CPSF30 there is likely only one copy of PAP brought to this subcomplex. Secondary to this, the authors present the first structure of Fip1 associating with CstF77 and provide biochemical validation of this model. Interestingly, the authors show that excess CstF77 leads to inhibition of in vitro polyadenylation of mPSF through the likely mutually exclusive association of Fip1 with 77 and PAP. Overall, the strengths of the study lie in high-resolution structural biology coupled with careful biochemical assays. My main concerns lie in the reduced novelty of the first half of this paper with the Hamilton and Tong study previously published. The main additional detail is whether one or two PAP molecules can associate with CPSF30/Fip1 and this seems somewhat incremental. There is a novelty to the observations that CstF77 can inhibit polyadenylation but this portion of the study is not as developed as it could be. If this portion of the paper were modestly expanded, I could be more convinced that it is worthy of publication. Several specific comments and suggestions are listed below.

1. The authors contend that their data support a model whereby a single copy of PAP is recruited to Fip1/CPSF30 in contrast to what was observed by Hamilton and Tong. It is important to note the experimental differences between the two systems: in this study, a GFP-PAP was used for in vitro binding, whereas Hamilton and Tong used only the catalytic module (1-524) of PAP; in this study, the authors use mPSF-PAP factors whereas in Hamilton and Tong full-length CPSF30 was used and Fip1 (79-200) was used. While I appreciate that experiments presented here appear to be done in a more 'complete' biochemical context, I am equally concerned that the use of GFP-PAP can somehow inhibit the ability of PAP to associate with Fip1/mPSF.

2. Does the inhibitory effect of CstF77 apply to other mRNA substrates beyond the one tested in this study?

3. Would supplementing the polyadenylation assays with the CstF complex (50/64/77) also cause inhibition, or is it unique to supplementation with isolated CstF77? If this were to be the case, how would this be interpreted?

*Reviewer #3 (Recommendations for the authors):*

1. The authors need to mention the role of FIP1 in alternative polyadenylation, and that it specifically promotes polyadenylation of mRNAs with U-rich stretches preceding the polyadenylation signal. The experiments of Shi et al. (Lackford et al., 2014), show that knockdown of Fip1 has a dramatic effect on alternative polyadenylation, resulting in skipping upstream polyA sites that have U-rich regions before the AAUAAA that bind Fip1. One potential interpretation of these experiments is that if FIp1 is limiting you can form an active polyadenylation complex that lacks Fip1, and complexes that lack Fip1 fail to cleave and polyadenylate at a subset of sites that require the U-rich elements for cleavage.

2. In interpreting the overall data, the authors should discuss possibilities of what actually happens in the cell when the cleavage complex forms on the nascent pre-mRNA. Their (and others) in vitro data on the properties of complexes not bound to substrate RNA describe the properties of complexes that may not be relevant to the functional complex that is bound to the pre-mRNA just prior to cleavage.

For example, they show CstF77 inhibits polyadenylation by interfering with the binding to Fip1. Since CstF and CPSF must interact during cleavage, this suggests that before cleavage (when all factors are bound to the substrate and RPPP6 is being recruited), PAP is likely not bound to the complex. After cleavage, either CstF may dissociate from the complex (since the 3' fragment of the RNA is bound to CstF) or alter its conformation to allow PAP to bind for polyadenylation. The authors might want to discuss these possibilities. This may be part of the reason there is a discrepancy in the literature on whether or not PAP is required for cleavage. If their model is correct PAP is likely not in the active cleavage complex (which contains CstF77), but might on the mPSF (or CPSF) that binds initially to the polyadenylation signal, but is lost when the active cleavage complex, which includes CstF forms.

3. They need to explicitly mention on page 5 (description of Figure 1) whether there are any differences between their structure and the structure of Tong and Hamilton of what seems to be the same complex.

4. They say (l. 138-139) "both Fip1 binding sites contribute to the integrity of stability of mPSF. Might be better to say that "either FIP1 binding site can contribute to the integrity of mPSF." The complexes with one FIp11 binding site look similar to those with 2 sites (wt) in Figure 1G.

5. In Figure 2A, it looks as if in mutant ZF4 or ZF5 polyadenylation is no longer processive. In the wild-type how long are the A-tails that accumulate? They should mention that. In the mutant AGUACA (I presume that is a mutant of the AAUAAA?) there is an addition of 3-6 A tails. It looks like the Y127E mutant is a little more active than the Y151E mutant (based on the length of the A tails). If that is reproducible that they could mention that.

6. The evidence that only one PAP is present in mPSF is convincing, and either FIP1 binding site seems by itself to be able to recruit PAP.

7. They describe and characterize the structure of a complex of the N-terminal 35 aa of Fip 1 with CstF77, adding a third interaction to the previously described interactions of CstF77 with Wdr33 and CPSF160. Since there are two copies of CstF77 in CstF, CstF can bind two molecules of Fip1, and they provide evidence that happens in mCSF, and is likely part of the mechanism by which mPSF and CstF interact during cleavage.

---

## [Author Response]

Essential revisions:1) Through thoughtful discussion, please address differences in approaches used to support the idea that a single copy of PAP exists in the complex in contrast to a previous publication. There are caveats to both studies and distinct experimental differences that should be highlighted to temper their overall narrative.2) Reviewers were particularly interested in the CstF77 data, specifically the interpretation. This is potentially important data to the field and should be discussed more in this study.Reviewer #1 (Recommendations for the authors):In the study by Muckenfuss et al., the authors provide two independent crystal structures of Fip1 in complex with CPSF30 or CstF77 as a basis for biochemical experiments exploring the function of Fip1 in polyadenylation. The structure of CPSF30/Fip1 is validated using polyadenylation assays and the authors conclude that while two molecules of Fip1 interact with a single molecule of CPSF30 there is likely only one copy of PAP brought to this subcomplex. Secondary to this, the authors present the first structure of Fip1 associating with CstF77 and provide biochemical validation of this model. Interestingly, the authors show that excess CstF77 leads to inhibition of in vitro polyadenylation of mPSF through the likely mutually exclusive association of Fip1 with 77 and PAP. Overall, the strengths of the study lie in high-resolution structural biology coupled with careful biochemical assays. My main concerns lie in the reduced novelty of the first half of this paper with the Hamilton and Tong study previously published. The main additional detail is whether one or two PAP molecules can associate with CPSF30/Fip1 and this seems somewhat incremental. There is a novelty to the observations that CstF77 can inhibit polyadenylation but this portion of the study is not as developed as it could be. If this portion of the paper were modestly expanded, I could be more convinced that it is worthy of publication. Several specific comments and suggestions are listed below.

We thank the Reviewer for the positive feedback and the constructive suggestions.

1. The authors contend that their data support a model whereby a single copy of PAP is recruited to Fip1/CPSF30 in contrast to what was observed by Hamilton and Tong. It is important to note the experimental differences between the two systems: in this study, a GFP-PAP was used for in vitro binding, whereas Hamilton and Tong used only the catalytic module (1-524) of PAP; in this study, the authors use mPSF-PAP factors whereas in Hamilton and Tong full-length CPSF30 was used and Fip1 (79-200) was used. While I appreciate that experiments presented here appear to be done in a more 'complete' biochemical context, I am equally concerned that the use of GFP-PAP can somehow inhibit the ability of PAP to associate with Fip1/mPSF.

We thank the Reviewer for the comment upon which we critically examined our experimental setup. While we indeed used GFP-PAP for our in vitro pull-down assays (Figure 1G, 2B, 4A), it is important to note that an untagged catalytic domain of human PAP (1-504) was used for copurification of a mPSF-PAP complex and its subsequent SEC-MALS analysis in isolation and in the presence of excess untagged PAP (Figure 2D). In fact, both mPSF and PAP are untagged in this analysis which ensures that any interference from affinity/epitope tags is eliminated.

Initially, we used untagged PAP for our in vitro pull-down analysis. However, due to the transient interaction of PAP with mPSF/hFip1 and the resulting weak bands on SDS-PAGE, we decided to use GFP-PAP for improved detection by in-gel fluorescence of the N-terminally fused GFP tag. For reference, a comparative pull-down analysis using untagged and tagged PAP with mPSF complexes is shown in Author response image 1. Equal amounts of PAP are precipitated by Strep-tagged mPSF complexes, independent whether it is fused to a GFP tag or not.

**Author response image 1. sa2fig1:** 

Furthermore, analysis of the three-dimensional structure of PAP reveals that the hFip1-binding site on PAP is located on the opposite face of the molecule relative to its N-terminus. Together with the fact that PAP is about twice the size (58 kDa) of the N-terminally fused GFP tag (28 kDa), it is unlikely that the tag would negatively affect its interaction with hFip1.Therefore, we believe that the different results observed in the two studies stem from the different compositions of the complexes (mPSF in our study vs. CPSF30-hFip1 in Hamilton and Tong) used to test for their interaction with PAP. Interestingly, Hamilton and Tong observe dimerization of CPSF30-hFip1 (70-200) complexes, which we do not observe with our mPSF complexes, an additional indication that isolated CPSF30-hFip1 complexes are able to establish additional interactions that are obstructed once the complex is integrated into the mPSF.

At present, we cannot exclude the possibility that mPSF complexes assemble with two PAP molecules in vitro at higher PAP concentration; however, we believe that this does not represent the predominant physiological assembly.

We made additional revisions to the manuscript, including the Discussion section, to elaborate on the observation of only one stably bound PAP molecule.

2. Does the inhibitory effect of CstF77 apply to other mRNA substrates beyond the one tested in this study?

We thank the Reviewer for raising this point. It is important to note that CstF64, rather than CstF77, is the principal RNA-binding component of the CstF complex that mediates interactions with G/U-rich downstream sequences through its RNA recognition motif (RRM) (Takagaki et al., 1997). CstF64, however, was not included in our analysis of CstF77-mediated inhibition of polyadenylation (and neither was CstF50) (Figure 4B, Figure 4 —figure supplement 1); furthermore, our use of pre-cleaved RNA substrates precludes the inclusion of downstream elements. As CstF77 does not possess an RRM and is not known to directly bind RNA (Bai et al., 2007; Yang et al., 2018), competition between CstF77 and PAP is expected to be mediated by protein-protein interactions rather than via the RNA substrate.

Following the suggestion of the Reviewer, we probed the inhibitory effect of CstF77 on polyadenylation with an alternative adenoviral L3 PAS-containing mRNA substrate (38 nt), that was previously shown to be processed in vitro by a recombinant human 3’ end processing complex (Boreikaite et al., 2022; Schmidt et al., 2022). Compared to SV40 mRNA, in which the PAS and cleavage site are separated by only 11 nt, these sites are separated by 20 nt in the L3 mRNA, which is within the normal range (15-21 nt) observed for mammalian pre-RNAs (Beaudoing et al., 2000; Gruber et al., 2016). We observe a similar degree of inhibition by CstF77 using the alternative adenoviral L3 mRNA substrate (Figure 4 —figure supplement 1B), suggesting that the inhibition is independent of the RNA substrate.

3. Would supplementing the polyadenylation assays with the CstF complex (50/64/77) also cause inhibition, or is it unique to supplementation with isolated CstF77? If this were to be the case, how would this be interpreted?

We thank the Reviewer for this intriguing question. hFip1 interacts with both CstF77 and CstF in our pull-down analysis (now: Figure 3 —figure supplement 3), indicating that the hFip1 binding site on CstF77 is not obstructed upon assembly of CstF77 into the CstF complex. As suggested by the Reviewer, we carried out polyadenylation assay using a recombinant holoCstF complex comprising CstF77, CstF64^1-198^, and CstF50 (now: Figure 4C) to demonstrate that holo-CstF causes inhibition of the polyadenylation reaction to a similar extent as isolated CstF77, suggesting that the inhibitory effect of CstF77 persists when it is assembled within CStF. As indicated, we added two additional figures to the manuscript (Figure 3 —figure supplement 3; Figure 4C) that further strengthen our findings on CstF-mediated inhibition of polyadenylation.

Reviewer #3 (Recommendations for the authors):1. The authors need to mention the role of FIP1 in alternative polyadenylation, and that it specifically promotes polyadenylation of mRNAs with U-rich stretches preceding the polyadenylation signal. The experiments of Shi et al. (Lackford et al., 2014), show that knockdown of Fip1 has a dramatic effect on alternative polyadenylation, resulting in skipping upstream polyA sites that have U-rich regions before the AAUAAA that bind Fip1. One potential interpretation of these experiments is that if FIp1 is limiting you can form an active polyadenylation complex that lacks Fip1, and complexes that lack Fip1 fail to cleave and polyadenylate at a subset of sites that require the U-rich elements for cleavage.

We thank the Reviewer for this comment. We now mention the role of hFip1 in alternative polyadenylation through modulating the selection of cleavage sites via its interaction with Urich sequence elements as well as cleavage factor Im in both the Introduction section of the manuscript, as well as in the Discussion.

2. In interpreting the overall data, the authors should discuss possibilities of what actually happens in the cell when the cleavage complex forms on the nascent pre-mRNA. Their (and others) in vitro data on the properties of complexes not bound to substrate RNA describe the properties of complexes that may not be relevant to the functional complex that is bound to the pre-mRNA just prior to cleavage.For example, they show CstF77 inhibits polyadenylation by interfering with the binding to Fip1. Since CstF and CPSF must interact during cleavage, this suggests that before cleavage (when all factors are bound to the substrate and RPPP6 is being recruited), PAP is likely not bound to the complex. After cleavage, either CstF may dissociate from the complex (since the 3' fragment of the RNA is bound to CstF) or alter its conformation to allow PAP to bind for polyadenylation. The authors might want to discuss these possibilities. This may be part of the reason there is a discrepancy in the literature on whether or not PAP is required for cleavage. If their model is correct PAP is likely not in the active cleavage complex (which contains CstF77), but might on the mPSF (or CPSF) that binds initially to the polyadenylation signal, but is lost when the active cleavage complex, which includes CstF forms.

We thank the Reviewer for raising these points. We have revised our Discussion section to expand on these ideas and discuss the possibilities, also with respect to the conflicting data in the literature on the requirement of PAP for pre-mRNA cleavage.

3. They need to explicitly mention on page 5 (description of Figure 1) whether there are any differences between their structure and the structure of Tong and Hamilton of what seems to be the same complex.

We note that although the structures are highly similar, they are not identical due to the different choices of protein construct boundaries. As a result, we have been able to resolve an additional 21 amino acid residues, mostly in the N- and C-terminal extensions of hFip1^CD^b. We have revised the relevant Results section to mention the similarities and differences between the two structures.

4. They say (l. 138-139) "both Fip1 binding sites contribute to the integrity of stability of mPSF. Might be better to say that "either FIP1 binding site can contribute to the integrity of mPSF." The complexes with one FIp11 binding site look similar to those with 2 sites (wt) in Figure 1G.

We thank the Reviewer for the comments. We have reworded the section as suggested. However, we respectfully disagree with the comment that the SDS-PAGE bands of GFP-PAP precipitated by single hFip1 and double hFip1 complexes (WT mPSF) are of similar intensities. Densitometric quantitation of the GFP fluorescence using ImageLab 6.1 software (see Author response table 1) reveals a marked reduction of both GFP-hFip1 and GFP-PAP band intensities upon mutation of either of the two hFip1 binding sites in CPSF30 (single hFip1 complexes) that consequently results in reduced recruitment of PAP. Nevertheless, so as not to over-interpret this data, we distinguish only between the presence or absence of interaction.

**Author response table 1. sa2table1:** Quantitation of co-precipitated GFP-hFip1 and GFP-PAP protein levels (GFP fluorescence detected at 473 nm), normalized against WT mPSF.

	wt	ZF4	ZF5
GFP-hFip	1	0.29	0.42
GFP-PAP	1	0.28	0.43

5. In Figure 2A, it looks as if in mutant ZF4 or ZF5 polyadenylation is no longer processive. In the wild-type how long are the A-tails that accumulate? They should mention that. In the mutant AGUACA (I presume that is a mutant of the AAUAAA?) there is an addition of 3-6 A tails. It looks like the Y127E mutant is a little more active than the Y151E mutant (based on the length of the A tails). If that is reproducible that they could mention that.

We thank the Reviewer for bringing up these points. We now added molecular weight markers to the gels showing polyadenylation assays. Furthermore, we specified more clearly that AGUACA is a mutant of the AAUAAA polyadenylation signal. As mPSF retains residual low affinity RNA binding activity in the absence of the canonical AAUAAA hexanucleotide (Clerici et al., 2017), residual polyadenylation activity can be observed for this RNA substrate. Although the slight difference in polyadenylation efficiencies of the Y127E and Y151E mutant is reproducible, we cannot exclude that this is due to the Y127E CPSF30 mutant (ZF4) still retaining some affinity for hFip1 (Figure 1E) as it has higher affinity for hFip1 than ZF5 does (Hamilton et al., 2020). Therefore, we refrained from further discussion of the results so as not to overinterpret the data.

6. The evidence that only one PAP is present in mPSF is convincing, and either FIP1 binding site seems by itself to be able to recruit PAP.

We thank the Reviewer for the positive feedback.